# u-µP: The Unit-Scaled Maximal Update Parametrization

**Charlie Blake** [* 1]  **Constantin Eichenberg** [* 2]  **Josef Dean** [1]  **Lukas Balles** [2]  **Luke Y. Prince** [1]  **Björn Deiseroth** [2]
**Andres Felipe Cruz-Salinas** [3 †]  **Carlo Luschi** [1 ‡]  **Samuel Weinbach** [2 ‡]  **Douglas Orr** [1]

## Abstract

The recent Maximal Update Parametrization (µP) enables the hyperparameters for small models to transfer directly to large ones, substantially reducing the cost of training by avoiding expensive sweeps at scale. We present a new scheme, u-µP, which improves upon µP by combining it with Unit Scaling, a method for designing models that makes them easy to train in low-precision. The two techniques have a natural affinity: µP ensures that the scale of activations is independent of model size, and Unit Scaling ensures that the starting-scale of these activations is one (along with weights and gradients). This synthesis opens the door to a simpler scheme, whose default values are near-optimal. This in turn facilitates a more efficient sweeping strategy, with u-µP models reaching a lower loss than comparable µP models and working out-of-the-box in FP8.

## 1. Introduction

Finding good hyperparameters (HPs) is critical to effective training, yet doing so for modern large language models (LLMs) is challenging. The huge scale of models and datasets makes performing multiple runs over a set of candidate HPs prohibitively expensive. The Maximal Update Parametrization (µP) aims to make HP values *consistent* across model sizes, allowing practitioners to sweep HPs on a small-scale *proxy* model and transfer them to a larger *target* model (Yang & Hu, 2021; Yang et al., 2022).

Whether µP has been used to train any recent leading LLMs has not been disclosed, though there are some indications of its use[1]. However, amongst the few LLMs for which

---
[*]Equal contribution  [1]Graphcore [2]Aleph Alpha [3]Cohere. [†]Work done while at Aleph Alpha. [‡]Supervisory role. Correspondence to: Charlie Blake <charlieb@graphcore.ai>, Constantin Eichenberg <constantin.eichenberg@aleph-alpha.com>.

Accepted to the Workshop on Advancing Neural Network Training at International Conference on Machine Learning (WANT@ICML 2024).

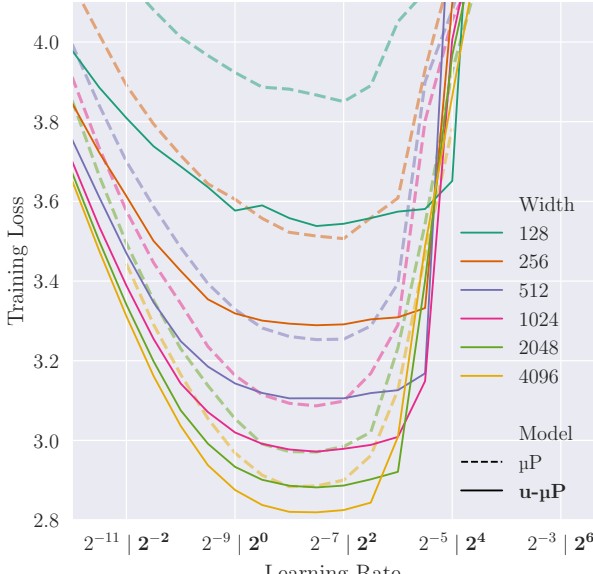

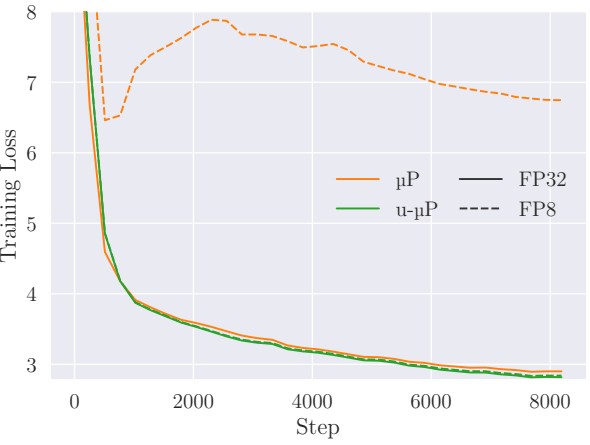

*Figure 1.* (Top) Using default HPs, u-µP models have lower losses and wider basins than µP models, while still transferring the optimal learning rate across width.

(Bottom) FP8 training via a simple cast. µP uses HPs found by random grid search here, whereas u-µP uses default HPs, only sweeping the LR. Width $= 4096$, $\eta = (2^{-7.5}, 2^{1.5})$ for (µP, u-µP).

comprehensive training details are known, µP has been a popular choice (Dey et al., 2023a;b; Liu et al., 2023; Hu et al., 2024). Indeed, it is the only principled method for hyperparameter transfer in the literature. Unfortunately users have not always found µP to provide effective transfer in practice (Almazrouei et al., 2023; Lingle, 2024), with a gap between the theory and its effective application. We propose an improved scheme: the Unit-Scaled Maximal Update Parametrization (u-µP). We specifically focus on LLM architectures in developing u-µP, as this is the domain in which µP has been used in practice.

Our method combines µP with another recent training innovation, Unit Scaling (Blake et al., 2023). Originally designed to facilitate simple low-precision training, Unit Scaling proposes the principle of unit variance at initialization for activations, weights and gradients. In doing so it relies on the same mechanism as µP: applying scalar multipliers to linear layers to counteract the effects of changing model size. This allows for a synthesis of the two methods and facilitates several improvements. u-µP retains the HP-transfer property of µP and the out-of-the-box FP8 training of Unit Scaling (Figure 1), but also simplifies the scaling rules (Table 1) and provides a more principled and interpretable set of HPs (Appendix D). The default values of these HPs (= 1) are near-optimal, giving lower losses than for µP (Figure 1) and opening up more efficient sweeping strategies (Figure 3).

## 2. Background

### 2.1. The Maximal Update Parametrization

As model sizes have grown, the behavior of neural networks during training in the limit of infinite width has become an active field of research. One of the most important results in this area is the classification of infinite-width limits under all possible *abc-parametrizations* (Yang & Hu, 2021). An abc-parametrization assumes that the training dynamics for a weight $W$ of a model are given by

$$W(t) = Aw(t), \quad w(0) \sim \mathcal{N}(0, B),$$
$$w(t+1) - w(t) = C\Phi_t(\nabla\mathcal{L}_0, ..., \nabla\mathcal{L}_t),$$

and provides a scheme for scaling the weight multiplier $A$, the initial variance $B$, and the learning rate $C$ with some exponent of the network width.

The authors prove that the only abc-parametrization (up to symmetry) that allows all model features to evolve non-trivially in the infinite-width limit without blowing up, is µP. We outline µP's parametrization rules in Table 1. This has significant implications for training at scale, as it suggests

that non-µP approaches will eventually be ineffective.

Another consequence of µP is that optimal hyperparameters transfer between different model sizes, a process known as µTransfer (Yang et al., 2022). This is not the case under the Standard Parametrization (SP) and has led to its adoption for LLM training. Yang et al. (2023) introduce a similar scheme for depth known as depth-µP[2], based on residual branch multipliers and learning rates (also see Table 1).

### 2.2. Unit Scaling

Unit Scaling is a paradigm to facilitate training with low-precision number formats. The use of these formats on modern AI hardware can bring substantial efficiency gains (Graphcore, 2022; Nvidia, 2022), but requires extra care to guarantee stable numerics. Existing efforts on FP8 training employ per-tensor scaling techniques, which are cumbersome to implement and incur additional overheads.

Unit Scaling takes a different approach, offering a paradigm for designing models that makes them inherently less likely to generate out-of-range values during training. It does so by ensuring that all activations, weights and gradients have *unit scale* (i.e. $\sigma = 1$) at initialization. As a consequence, unit scaled models can be trained in low-precision via a simple cast operation.

This is achieved through the application of scalar multipliers to every operation in the forward and backward passes, counteracting the scale-changing effect of operations like matrix multiplications. For example, the rule when multiplying a vector by a square matrix of length $d$ is to re-scale the result by $1/\sqrt{d}$. Where different scales are required in the forward and backward passes, in certain cases Unit Scaling is able to leverage the *cut-edge rule* to satisfy this requirement without breaking the chain rule (see Appendix A.3 for further details).

### 2.3. Low-precision training

While full-precision (FP32) floating-point arithmetic has been the default in the machine learning community for many years, the ever-growing scale of models and datasets has led to a push towards lower-precision formats. The use of FP16 formats has seen uptake in large-scale Transformer training, but requires extra care to guarantee stable numerics. Certain quantities, such as optimizer states, are routinely kept in full precision and loss scaling techniques (Micikevicius et al., 2018; Kuchaiev et al., 2018) are employed to keep gradients in the representable range of the FP16 format. More recently, BFLOAT16 training has mitigated these scaling issues for 16-bit formats. However, no such alternative is available for narrower formats, such as FP8.

---

[1]The GPT-4 technical report (OpenAI, 2023) includes Yang et al. (2022) in its references, though makes no explicit mention of its use. The multipliers present in Grok (xAI, 2024) may also suggest the use of µP.

[2]When we refer to µP elsewhere in the paper we implicitly assume the modifications introduced in depth-µP.

*Table 1.* **The µP and u-µP schemes (expressed in absolute terms)**

| | Weight type | Input | Output | Hidden | Residual |
|---|---|---|---|---|---|
| **µP** | Init. Var. | $\sigma_{\text{init}}^2$ | $\sigma_{\text{init}}^2$ | $\sigma_{\text{init}}^2 \frac{\text{base-fan-in}}{\text{fan-in}}$ | — |
| | Multiplier | 1 | $\frac{\text{base-fan-in}}{\text{fan-in}}$ | 1 | $\sqrt{\frac{\text{base-depth}}{\text{depth}}}$ * |
| | Adam LR Mult. | 1 | 1 | $\frac{\text{base-fan-in}}{\text{fan-in}}$ | $\sqrt{\frac{\text{base-depth}}{\text{depth}}}$ |
| **u-µP** | Init. Var. | 1 | 1 | 1 | — |
| | Multiplier | 1 | $\frac{1}{\text{fan-in}}$ † | $\frac{1}{\sqrt{\text{fan-in}}}$ | $\frac{1}{\sqrt{\text{depth}}}$ * |
| | Adam LR Mult. | $\frac{1}{\sqrt{\text{fan-out}}}$ | 1 | $\frac{1}{\sqrt{\text{fan-in}}}$ | $\frac{1}{\sqrt{\text{depth}}}$ |
| **µP** | Associated HPs | (8): $\eta, \hat{\eta}_{\text{emb}}$, base-width, base-depth, $\sigma_{\text{init}}, \alpha_{\text{emb}}, \alpha_{\text{attn}}, \alpha_{\text{output}}$ | | | |
| **u-µP** | Associated HPs | (6): $\eta, \alpha_{\text{residual}}, \alpha_{\text{residual-attn-ratio}}, \alpha_{\text{ffn-act}}, \alpha_{\text{attn}}, \alpha_{\text{output}}$ | | | |

*Residual mults are applied to the end of each branch, rather than the output of linear layers.

†To maintain unit scale we apply $^1\!/\!\sqrt{\text{fan-in}}$ scaling in the backward pass (see Appendix A.3).

Recently released AI accelerators have introduced native hardware support for FP8 arithmetic (Graphcore, 2022; Nvidia, 2022). However, the further reduced precision and range of these formats introduces additional numerical challenges which have not been addressed conclusively by the literature.

While FP16 arithmetic can be used throughout the complete forward-backward pass through the model, efforts on FP8 training have focused on performing the computationally most expensive operations, matrix multiplications, in FP8. This means that input tensors are cast to an FP8 format prior to a matrix multiplication, the result of which is produced in a higher-precision format again.

Micikevicius et al. (2022) propose two different FP8 formats for deep learning, assigning different numbers of bits to the exponent and the mantissa, respectively. The E4M3 format uses 4 bits for the exponent and 3 bits for the mantissa, whereas the E5M2 format uses 5 bits for the exponent, prioritizing range over relative precision. (The maximum representable value of E5M2 is 57344 compared to 448 for E4M3.) The authors recommend to cast activations and weights to E4M3 and activation gradients (computed during the backward pass) to E5M2.

Finally, existing attempts at FP8 training employ per-tensor scaling techniques (Micikevicius et al., 2022). A scalar factor is extracted prior to casting a tensor to FP8 and multiplied back onto the output. In pseudo-code that reads:

```
a = scale(A)
b = scale(B)
A = to_fp8(A / a)
B = to_fp8(B / b)
C = (a * b) * matmul(A, B)
```

where we assume that `matmul` takes inputs in FP8 and

directly produces the output in higher precision. An obvious choice for the scaling factor is to rescale the maximum absolute value of a tensor to the maximum value representable by the FP8 format (Micikevicius et al., 2022). However, this requires computing "absmax" prior to performing the matrix multiplication, which hinders an efficient implementation. To circumvent that, a so-called delayed scaling technique has been proposed (NVIDIA, 2024), which tracks the scale of each tensor over time, allowing rescaling and matrix multiplication to be implemented in an efficient fused kernel.

## 3. Combining µP with Unit Scaling

In this section we derive our new u-µP scheme. The final set of scaling rules is given in Table 1.

### 3.1. From relative to absolute scaling

We seek to implement the rules specified by µP in a way that maintains unit scale for all tensors at initialization. Our first challenge is that whereas Unit Scaling provides *absolute* initialization values and multipliers, µP only specifies a *parametrization*. Yang et al. (2022) define a parametrization as 'a rule for how to change hyperparameters when the widths of a neural network change, [not] how to set the hyperparameters for any specific width'. To satisfy Unit Scaling we require our u-µP scheme to provide absolute scaling rules[3], so our first step is to derive an absolute-value implementation of µP. We do this by combining µP's 'base shapes' and initialization HPs with its standard scaling rules, resulting in Table 1.

The non-unit initialization of µP here violates Unit Scaling. However, due to the symmetry of abc-parametrizations (see

---

[3]In this sense u-µP is not strictly a *parametrization*, but we retain the term for simplicity.

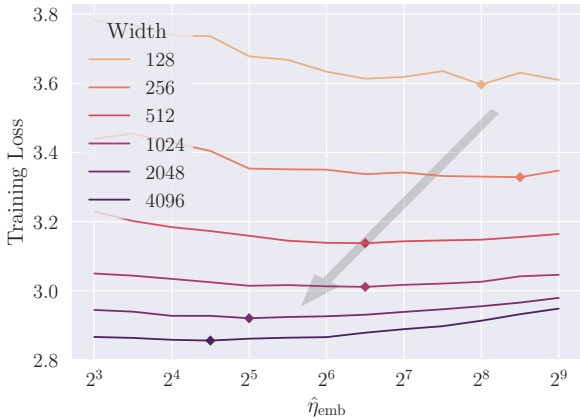

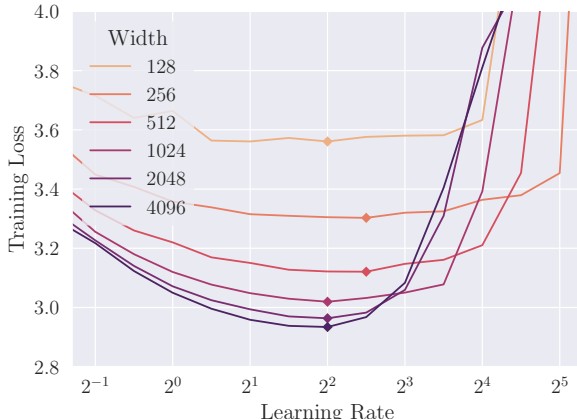

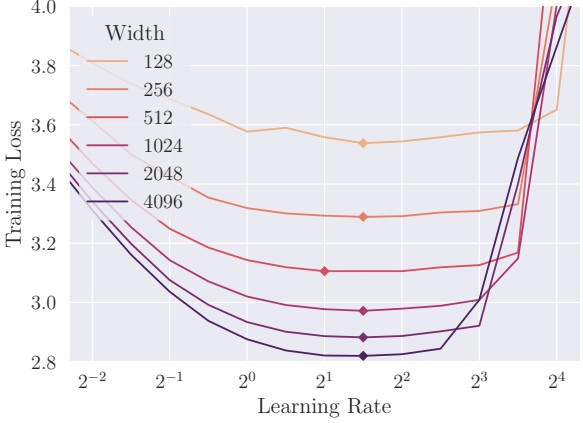

*Figure 2.* (Top) Transfer of the input LR multiplier over width when using µP.

(Middle) LR transfer over width using modified u-µP *without* our $1/\sqrt{\text{fan-out}}$ input LR scaling rule. Without our proposed rule, u-µP would have worse transfer and absolute loss at large width.

(Bottom) LR transfer over width using standard u-µP *with* our $1/\sqrt{\text{fan-out}}$ input LR scaling rule. (Reproduced from Figure 1 for comparison.)

Section 2.1), we can move scales between the initialization, multiplier and learning rate to attain any desired initialization, choosing variance of 1 to give unit-scaled weights. This results in the intermediate scheme shown in Table 2, Appendix A.1.

We impose two further changes to simplify the scheme and facilitate better transfer: dropping the 'base shapes' and $\sigma_{\text{init}}$ HPs, and changing the input LR rule to $1/\sqrt{\text{fan-out}}$ (these changes are justified in Appendix A.2 and Section 3.2). This results in the u-µP scheme given in Table 1.

The input and hidden multiplier rules that result from this scheme are exactly those specified by Unit Scaling (a reflection of its close relationship with µP). The u-µP output multiplier breaks Unit Scaling temporarily, but this is not a problem in practice as output mis-scaling does not propagate in the forward pass and output gradients can be corrected by $1/\sqrt{\text{fan-in}}$ scaling thanks to the *cut-edge rule* (see Appendix A.3). The only other point at which Unit Scaling is violated is the $1/d_{\text{head}}$ attention scaling, but this again is temporary. Otherwise, our resulting u-µP scheme has unit-scale weights, gradients and activations, while satisfying µP.

### 3.2. A new embedding scaling rule

Our only fundamental deviation from µP in terms of scaling rules is changing the embedding LR multiplier ($\hat{\eta}_{\text{emb}}$) from not being scaled to being scaled by $\sqrt{1/\text{fan-out}}$ (i.e. $\sqrt{1/\text{width}}$). This is based on the empirical observations shown in Figure 2, where the µP embedding learning rate parameter does not transfer well with width.

The top plot demonstrates that the best $\hat{\eta}_{\text{emb}}$ at e.g. the 256 width does not transfer to any subsequent width, with a clear drift towards smaller multiplier values. This drift is approximately corrected by scaling the embedding learning rate by $\sqrt{1/\text{width}}$, which justifies our change to the scaling rule provided by µP. The equivalent experiments for u-µP are shown in Figure 6, which exhibits the same problem until our correction is introduced.

This adjustment makes a significant difference in practice, as the original scaling rule causes performance to degrade as width increases. The middle and bottom plots demonstrate this for u-µP without and with our $\hat{\eta}_{\text{emb}}$ scaling correction. Note that this phenomenon of poor embedding LR transfer only has a minor effect on the *transfer* of the global LR, but significantly degrades the overall *loss* as we scale to larger models. We have observed similar issues for µP models, where sweeping an embedding LR multiplier can also impair the performance of the scaled-up model. We leave a potential theoretical explanation of why the original µP rules do not give transfer here to future work. It may be the case that at much larger scales the original rule is more appropriate, though this is unlikely to be of practical use.

### 3.3. A maximally independent set of hyperparameters

The original set of µP HPs outlined in Table 1 is based on common usage (i.e. those used for training LLMs in the literature, though the particular HPs for each model vary) and there is complex interplay and redundancy between some HPs (see Appendix E.1), in the sense that they effectively control the same scale in the model. We aim for a set of HPs that is maximally expressive while having no such redundancy. Besides the global learning rate parameter $\eta$, we end up with the following HPs:

- Residual branch multipliers: $\alpha_{\text{residual}}$ and ratio $\alpha_{\text{residual-attn-ratio}}$ (see eq. (19)).

- Multipliers before non-homogeneous operations: $\alpha_{\text{attn}}$, $\alpha_{\text{ffn-act}}$, $\alpha_{\text{loss-softmax}}$ (see eq. (25)).

This set of HPs, although smaller than the set of original µP HPs, generates a rich set of training dynamics. Furthermore, it has no redundancy and each multiplier controls a meaningful scale in the model. For a more detailed explanation, see Appendix D.

## 4. Additional Unit-Scaled Ops

For the sake of our experiments, a set of unit-scaled ops are required for Llama-style (Touvron et al., 2023) transformer architectures. We are able to utilize many of the ops given in the Unit Scaling paper (Blake et al., 2023). However, this selection lacks some of the features required to implement modern LLMs. To address this, in this section we outline a series of new unit-scaled ops for each of our required architectural features.

The presentation here is derived from that of the Unit Scaling Compendium given in Blake et al. (2023, appendix G). This makes reference to the factors $\alpha, \beta_1, \dots, \beta_k$. $\alpha$ is the output scaling factor in the forward pass, and $\beta_i$ are the scaling factors for the gradient of the op's inputs in the backward pass. For each op, a value or rule is provided for determining the required mult to ensure unit scale. The correct value for these multipliers is derived by analyzing the scaling behavior of each op, given some reasonable distributional assumptions about the input and incoming gradient tensors.

We provide a summary of these results in Table 4, which can be seen as an extension of Table A.2 in the Unit Scaling paper.

**Unit-scaled dot-product attention**  The Unit Scaling paper considers the attention layer scaling in terms of its separate components: the various matmul operations and the internal softmax. Linear operations are scaled using the standard rule, and the softmax scaling is given a $\alpha = \beta = s$ factor.

From an implementation perspective, the self-attention layer is more typically broken down into weight-matmuls and a fused scale-dot-product attention operation. This is the case we handle here, accounting for three complicating factors not considered in the Unit Scaling paper:

1. As we use a decoder-style transformer in our experiments, our softmax operation has a causal mask applied to its input.

2. We follow the µP guidance of using $1/d_{head}$ scaling in our self-attention layer, rather than the usual $1/\sqrt{d_{head}}$.

3. We place a $\alpha_{\text{attn}}$ multiplier immediately before the softmax, which is an HP that users may tune.

As a result our dot-product attention takes the form:

$$\text{attention}(q, k, v) = \text{softmax}\left(\alpha_{\text{attn}} \cdot \frac{q \cdot k^\top}{d_{\text{head}}^{-1}} \cdot c_{\text{mask}}\right) \cdot v$$

The addition of an HP before the softmax introduces an additional challenge for Unit Scaling, as our scaling multipliers will need to account for this value when preserving unit scale.

This operation is sufficiently complex that we found an empirical model of its scale to be more accurate than any mathematically-derived rule (future work may consider justifying our model mathematically). We find that the scale of dot-product attention is approximately

$$\sigma(\text{attention}(q, k, v)) =$$
$$\text{log\_interpolate}\left(\frac{1}{1 + \frac{4d_{\text{head}}}{\alpha_{\text{attn}}^2}}, 1, \sqrt{\frac{\log(s)}{s}}\right)$$

where

$$\text{log\_interpolate}(\alpha, b_{\text{upper}}, b_{\text{lower}}) =$$
$$e^{\alpha \log(b_{\text{upper}}) + (1-\alpha)\log(b_{\text{lower}})}.$$

The corresponding scaling rule is therefore to divide by this factor in both the forward and backward pass, as outlined in Table 4.

**SwiGLU FFN**  Llama uses a SwiGLU (Shazeer, 2020) layer for its FFN, which introduces two new operations for us to unit-scale: a SiLU (Yu & Su, 2019) (a.k.a. swish (Ramachandran et al., 2018)) operation and an element-wise multiplication. We take a similar approach to our dot-product attention, and consider unit-scaling the following fused operation:

$$\text{gated\_silu}(x_{\text{in}}, x_{\text{gate}}) =$$
$$x_{\text{in}} \odot x_{\text{gate}} \odot \text{sigmoid}(\alpha_{\text{ffn-act}} \, x_{\text{gate}})$$

For the surrounding weight-matmuls we follow the standard Unit Scaling rules.

Again, we use an empirical model of the scale of this op, which is surprisingly similar to the dot-product attention model:

$$\sigma(\,\text{gated\_silu}(x_{\text{in}}, x_{\text{gate}}))$$
$$= \text{log\_interpolate}\left(\frac{1}{1+\frac{1}{\alpha_{\text{ffn-act}}^2}}, \frac{1}{\sqrt{2}}, \frac{1}{2}\right),$$

dividing through by this factor to get our scaling rule.

**Residual layers**  Our implementation of residual layers for u-μP is more complex than other operations, as adjustments are required to:

1. Make pre-norm residual networks support Unit Scaling (see Appendix C).

2. Introduce our new, principled residual HPs (see Appendix D).

Our residual layer scheme is presented in full in D.1.2. For readers interested in our justification for this, see the sections noted above.

As mentioned in Appendix A.3, we also follow the example of Unit Scaling and delay the application of our residual multiplier in the backward pass to the base of the branch (see Blake et al. (2023), Figure 3c). This does not change the model, and enables unit scale to be maintained on the residual branch regardless of the value of the multiplier.

**RoPE embeddings**  We also require a unit-scaled implementation of Rotary Position Embeddings (RoPE (Su et al., 2024)), which are applied just before the scaled dot-product attention operation. As RoPE essentially consists of pairwise rotations of elements by different degrees, we observe no meaningful scale-change as a result of it's application, and hence leave it unchanged.

**RMSNorm**  Following Lingle (2024) we opt to use a non-trainable version of RMSNorm (Zhang & Sennrich, 2019), in order to facilitate better transfer. As a result, we also leave this operation unchanged. Were a trainable RMSNorm to be used, the recipe would follow closely that of the LayerNorm presented in the original Unit Scaling Compendium.

**Scale constraints**  One final, minor deviation from the scheme outlined in the Unit Scaling paper is the way in which we apply scale constraints (see their Section 5.2). The essence of scale constraints is that for perfect unit scaling, sometimes the ideal scale for the forward pass differs from those in the backward pass. In some special cases (e.g. at

the ends of the network) the use of different scales can be valid, but in the general case a single scale must be agreed upon. The solution in the Unit Scaling paper is to use the geometric mean of the forward and backward scales.

We propose instead to simply use the forward scale over the backward scale(s) in these cases. We do so for the following reasons:

1. For these architectures we find empirically that where there is a disparity in ideal forward and backward scales, it is not large.

2. By taking the forward scale, we can ensure strict unit-scale in the forward pass.

The value of the latter point is in terms of what it means for the interpretation of our u-μP multiplier HPs. Consider the $\alpha_{\text{ffn-act}}$ multiplier; with strict unit scale we can say that the standard deviation of activations immediately before this multiplier is 1. Therefore the standard deviation immediately after is $\alpha_{\text{ffn-act}}$. As this multiplier is (by design) the last operation before the ffn activation function, we can say that the interpretation of $\alpha_{\text{ffn-act}}$ is simply to set the input standard deviation to the FFN's activation function. Similar arguments can be made for other u-μP multiplier HPs. This interpretation only holds because we use the forward-scale in our constraints.

## 5. Experiments

We provide empirical results to support our claim that u-μP's HPs facilitate more efficient sweeping than μP while retaining good HP transfer, and demonstrate that it provides out-of-the-box FP8 training without dynamic scaling. Our experiments use the Llama (Touvron et al., 2023) architecture, trained on WikiText-103 (Merity et al., 2017) and evaluated using final training cross-entropy loss.

### 5.1. Experimental details

**Training and evaluation**  Our experiments are all in the setting of autoregressive language model training, a domain that has proven a useful testing ground for general machine learning techniques in the model-capacity constrained regime (the under-fitting regime described by Belkin et al. (2019)). Our evaluation metric is final training cross-entropy loss. This has the benefits of low variance and of separating the concerns of downstream training and regularization, which are not in scope for this work. This follows the precedent of Yang et al. (2022) who also report training loss. As our model training is not in the over-fitting regime, we expect training loss to track validation (and have seen so empirically). Default training settings are given in Table 6, with further experimental details in Appendix E.

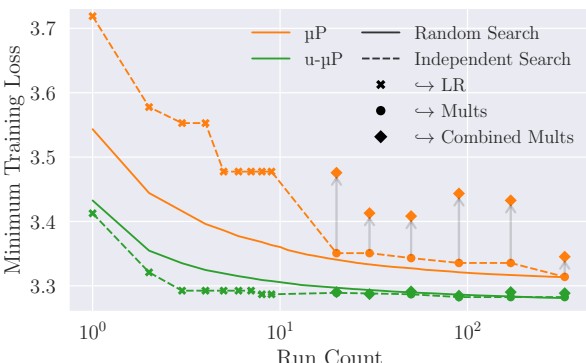

*Figure 3.* A comparison of random versus independent HP search strategies on μP and u-μP, showing that u-μP parameters are more amenable to independent search, and that combining the results of independent searches of μP mults harms performance.

**FP8 scheme** We demonstrate that u-μP enables a simple scheme for FP8 training, without the need for cumbersome per-tensor scaling. Each linear module in a model induces three matrix-matrix products: one during the forward pass to compute the output and two during the backward pass, computing gradients w.r.t. the weights and the inputs, respectively. The tensors participating in these matrix-matrix products are the input activations, the weights and the gradients w.r.t. the output activations. Unit scaling ensures unit scale for all three tensors at initialization.

*Empirically*, the scales of these tensors do not drift too much during training with the exception of the input tensors to the FFN and self-attention final projections, which grow considerably (see Appendix E.3, in particular Figure 9). This is consistent across different HP settings (see Figure 10).

Based on this observation, we test a simple scheme for FP8 training. For every matrix multiplication, we cast the input, weight and grad-output tensors to E4M3, with the exception of the inputs to FFN and self-attention final projections, which are cast to E5M2 to accommodate their growing scale. The output of each matrix multiplication is produced directly in the higher-precision format (FP32 in our case). No loss scaling or per-tensor scaling is applied.

Note that we conducted our experiments with simulated FP8 numerics, quantizing inputs to a matrix multiplication as if they were cast to FP8, while allowing the multiplication to be computed on hardware without native FP8 support.

### 5.2. Hyperparameter search

The set of u-μP HPs has been chosen such that their effects should be independent (see Appendix E.1), with their default ($= 1$) values also providing unit-scale inputs to associated functions. To test the benefit of this, we evaluate two sweeping strategies on μP and u-μP. First, a random

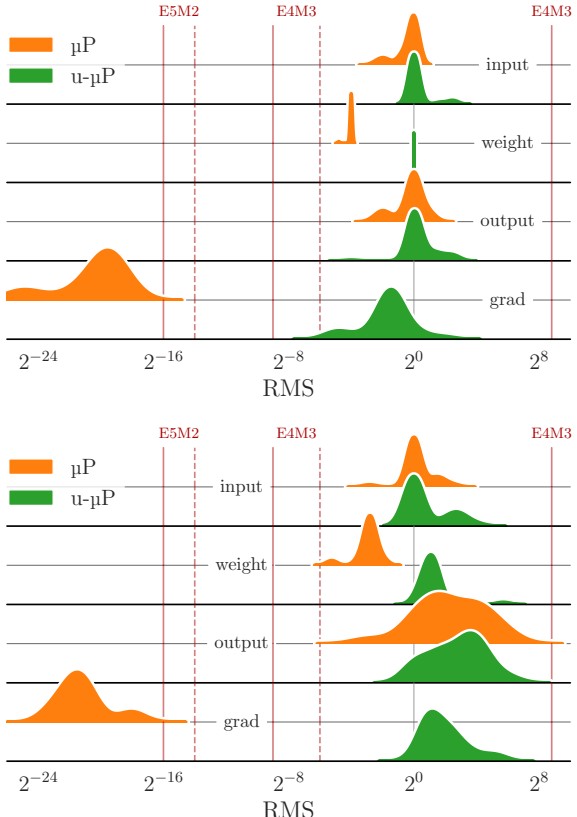

*Figure 4.* Per-tensor RMS $= \sqrt{\sigma^2 + \mu^2}$ across u-μP and μP models at initialization (top) and after training (bottom). u-μP tensors have RMS that starts close to 1 and remains within E4M3 range at the end of training. Dashed and solid red lines show each format's min. normal and subnormal values.

grid search following (Yang et al., 2022; Dey et al., 2023a;b; Hu et al., 2024), which should perform well in the presence of HP interactions. Second, an independent search which performs a line search for each HP in parallel, then combines the optima. Figure 3 shows the results: u-μP is amenable to both methods, while μP does not perform well under independent search. Hence u-μP enables more efficient sweeping strategies than μP, with near-optimal loss after just the LR portion of the search.

### 5.3. Hyperparameter transfer

Figure 1 (top) demonstrates that u-μP retains the key learning rate transfer across width property of μP. Moreover, using the default mult values, u-μP models are able attain the optimal loss for μP models of twice the width. We also demonstrate that the learning rate of u-μP transfers across the number of training steps, batch size and depth in Figure 5, similarly to μP but again with better loss using default HP values. Figures 2 and 6 show that u-μP's change to the input scaling rule improves width transfer and absolute loss.

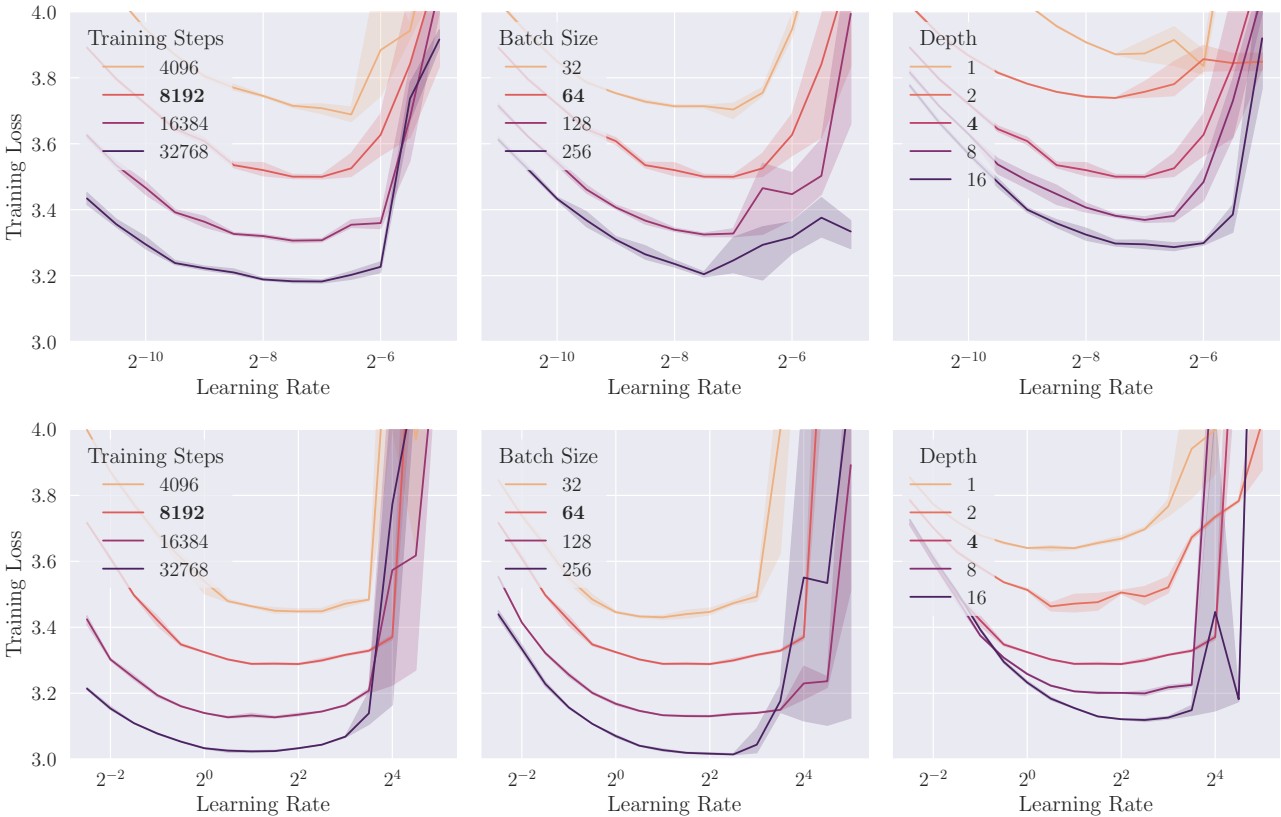

*Figure 5.* Learning rate transfer for µP (top) and u-µP (bottom), over training steps, batch size and depth. See Figure 1 (top) for transfer over width. The **default** shape parameter for other panels is shown in bold. The shaded area shows the 95% confidence interval for the mean.

## 5.4. Numerical properties

u-µP's scaling rules and multipliers are designed to maintain unit variance of activations, weights and gradients at initialization. Hence, these tensors are initialized well within the range of FP8 formats and, empirically, do not drift out of range during training (see Figure 4). Based on this, we propose and test a simple proof-of-concept scheme for FP8 training with u-µP. For every linear module, we cast input, weight and grad-output tensors to FP8 (without any scaling) prior to the matrix multiplication. Extra care is required for the inputs to FFN and self-attention final projections; the details of our scheme may be found in Section 5.1. Figure 1 (top) and Figure 8 show that FP8 u-µP converges close to the baseline value, while µP fails to train. Comprehensive numerics experiments can be found in Appendix E.3.

## 6. Related work

The Tensor Programs series of papers (Yang, 2019; 2020a; Yang & Littwin, 2021; Yang, 2020b; Yang & Hu, 2021; Yang & Littwin, 2023; Yang et al., 2022; 2023) introduce the mathematical framework from which µP is derived. Dey

et al. (2023a;b); Liu et al. (2023); Hu et al. (2024) employ µP to derive the HPs for LLM training, with Lingle (2024) exploring the transfer properties of µP variants. Blake et al. (2023) introduce Unit Scaling, which has similarities to Klambauer et al. (2017); Yuan & Zhu (2022) in its activation function scaling, and to Glorot & Bengio (2010); He et al. (2015) in its approach to initialization.

## 7. Conclusions

We demonstrate the effectiveness of our improved scheme, which applies the principles of Unit Scaling to µP to form u-µP. We retain the HP transfer property of µP and benefit from the simple low-precision training brought by Unit Scaling. This allows FP8 via a simple cast, bringing substantial efficiency gains. Our u-µP HP scheme is more principled than that used for µP, leading to a simpler sweeping strategy. Indeed the default multipliers associated with u-µP models are near-optimal in our experiments, reaching a lower loss than heavily-tuned µP models. The combination of these benefits indicates that u-µP can be a valuable component in the simple, stable and efficient training of LLMs.

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

# Contents

# A. From µP and Unit scaling to u-µP

Although u-µP provides the benefits of both µP and u-µP and is faithful to almost all of their rules, it includes some specific changes that deviate from the parent schemes. In this section we give additional details about u-µP as well as highlighting and explaining the changes from µP and Unit Scaling.

## A.1. An intermediate scheme

The first step from µP to u-µP is enabled by the abc-symmetry (see Lemma J.1 in (Yang et al., 2022)). The initial hidden weight variance $\text{base-fan-in}/\text{fan-in}$ from Table 1 can be replaced by 1 if we change at the same time:

- Multiplier: $1 \to \sqrt{\frac{\text{base-fan-in}}{\text{fan-in}}}$

- Adam LR Mult.: $\frac{\text{base-fan-in}}{\text{fan-in}} \to \sqrt{\frac{\text{base-fan-in}}{\text{fan-in}}}$

Together with setting $\sigma_{\text{init}} = 1$ we get the intermediate scheme shown in Table 2.

*Table 2.* **µP scheme after abc symmetry and normalized init. variance**

|  | Weight type | Input | Output | Hidden | Residual |
|---|---|---|---|---|---|
|  | Init. Var. | 1 | 1 | 1 | — |
| **µP** | Multiplier | 1 | $\frac{\text{base-fan-in}}{\text{fan-in}}$ | $\sqrt{\frac{\text{base-fan-in}}{\text{fan-in}}}$ | $\sqrt{\frac{\text{base-depth}}{\text{depth}}}$ |
|  | Adam LR Mult. | 1 | 1 | $\sqrt{\frac{\text{base-fan-in}}{\text{fan-in}}}$ | $\sqrt{\frac{\text{base-depth}}{\text{depth}}}$ |

## A.2. Dropping base shapes

The next step in our transition from the µP scheme in Table 2 to our u-µP scheme in Table 1 is the removal of the 'base shapes' HPs, base-fan-in and base-depth. It should be noted that although we refer to base-fan-in and base-depth in the same way as the multiplier HPs, these are not intended to be swept, but rather chosen according to taste. The stated purpose of the 'base shapes' in Yang et al. (2022) is to enable a model to behave unchanged at a particular size, yet still scale according to µP rules as the model-size changes.

We argue that this is not necessary or desirable for u-µP based on the following:

1. Though in some cases it may be useful to align a µP model to some 'base' model in the manner described above, all uses of µP in the literature simply set the base shapes to those of a small model and perform an HP sweep on top of it. This sweep alters the dynamics of the base model before scaling-up, so there is little sense in which the base shapes maintain the behavior of some smaller model as we scale. At best the base shapes can be seen as giving a useful starting point to the HP sweep.

2. In contrast u-µP offers a *principled* approach to the base dynamics of the model: that all tensor-scales should have unit variance. This can be seen as an alternative to the 'base shapes' approach, which substitutes this principle for alignment with non-µP models at a particular scale.

3. The effect of base shapes on the weight multipliers is a constant. Our set of u-µP multipliers is able to express any set of constant weight multipliers in the network, so our HP sweep is in effect testing what might happen were we to introduce base shapes into u-µP.

4. The effect of base shapes on the LR multipliers is again a constant factor, which is applied to all hidden weights for width, and residual weights for depth. This applies to the large majority of weights in the model in a similar fashion, making it closely linked to a global shift in the LR. As the LR is swept anyway, we conjecture that the effect of base shapes on the model's LRs is minimal.

5. Finally, we argue for simplicity. If we can remove these additional HPs without detriment, then our scaling rules become much simpler: each depends on a single factor.

With this change implemented, our scaling rules become those shown in Table 3.

*Table 3.* **µP scheme after abc symmetry, normalized init. variance and dropping of base shapes**

|  | Weight type | Input | Output | Hidden | Residual |
|---|---|---|---|---|---|
| **µP** | Init. Var. | 1 | 1 | 1 | — |
|  | Multiplier | 1 | $\frac{1}{\text{fan-in}}$ | $\frac{1}{\sqrt{\text{fan-in}}}$ | $\frac{1}{\sqrt{\text{depth}}}$ |
|  | Adam LR Mult. | 1 | 1 | $\frac{1}{\sqrt{\text{fan-in}}}$ | $\frac{1}{\sqrt{\text{depth}}}$ |

### A.3. Forward multipliers vs. backward multipliers

Table 1 shows the scaling rules for weight multipliers in u-µP. If we are more precise, these values are actually post-op forward multipliers, i.e. a multiplier $\alpha$ in a linear layer $f$ gets applied as $h \mapsto \alpha \cdot f(h)$. This is because for low-precision training, we cast the input $h$ to FP8 in the forward pass and a pre-op multiplication $\alpha \cdot h$ might go out of FP8 range if the multiplier is too small or too large. Conversely, we want $\alpha$ to act as a pre-op multiplier in the backward pass, so we have $g_h \mapsto \alpha \cdot \text{grad}_f(g_h)$ because in this case the gradient $g_h$ gets cast to FP8. The same logic applies to residual multipliers, with the branch multiplier implemented after the residual path in the forward pass, and before the residual path in the backward pass.

This is straightforward when the forward and backward multipliers are the same, but nevertheless crucial to implement correctly in order to enable stable FP8 forward and backward passes. However, there are two instances in u-µP where forward and backward multiplier are different:

1. **Weight gradients.** While model weights under Unit Scaling are correctly scaled in the forward pass, the weight gradient computation involves a summation over the batch size $b$, hence from a Unit Scaling perspective we need to apply a scaling in $b$ to keep the weight gradient unit scaled. We choose the scaling

$$g_w \mapsto b^{-\frac{1}{2}} g_w,$$

   that works well in practice. The discrepancy between forward and backward computation can be easily resolved post-hoc in the optimizer function that calculates the weight update from its gradients. In the case of Adam, no adjustment is needed because of its scale invariance property.

2. **Readout layer**. In order to satisfy µP and prevent logits from blowing up as the network width increases, the readout layer has a forward multiplier of $1/\text{fan-in}$. This contradicts Unit Scaling, but is not too problematic in the forward pass since we are only under-scaled at the first step of training and then have logits of order 1. Also, this happens at the very end of the network and has no significant effect on any subsequent operations, since the loss computation usually stays in higher precision. In the backward pass however, using this multiplier leads to all gradients in the model becoming under-scaled throughout training. The fix is to use the backward multiplier $\sqrt{1/\text{fan-in}}$ instead. Again, this produces "mathematically incorrect" gradients. Because the readout layer is not on a residual branch and the backward pass is linear, we can again easily compensate for this static factor in the optimizer.

A unified explanation for why different forward and backward scale are admissible for weights and readout layer, but not for operations on residual branches, is given by the cut-edge-rule (see Section 5.1 in (Blake et al., 2023)).

### A.4. Additional figures for the new embedding scaling rule

Figure 6 provides additional results showing transfer of the input LR multiplier $\hat{\eta}_{\text{emb}}$ for u-µP models (whereas the Figure 2 plot in the body of the paper shows this for regular µP models).

## B. Additions to the Unit Scaling compendium

We provide a table summarizing our new unit-scaled ops in Table 4, to accompany the definitions set out in Section 4.

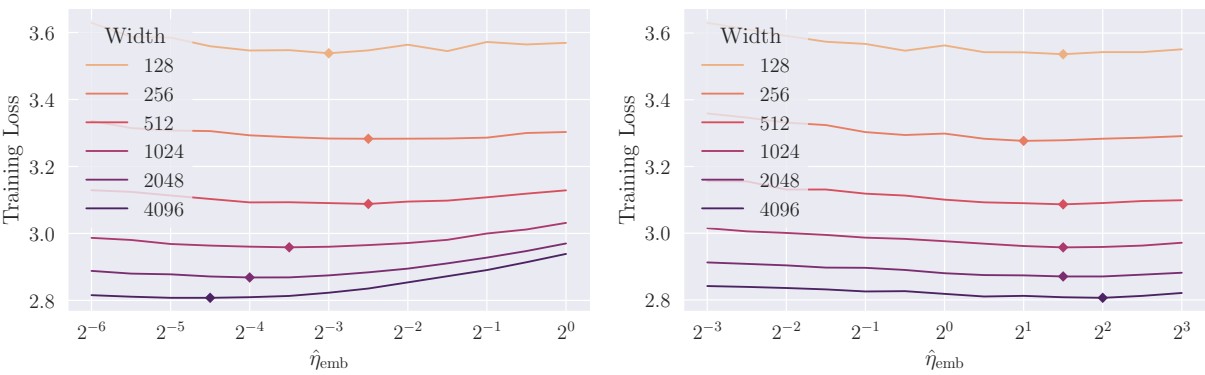

*Figure 6.* Transfer of the input LR multiplier $\hat{\eta}_{\text{emb}}$ over width. (Left) Modified u-µP *without* our $1/\sqrt{\text{fan-out}}$ input LR scaling rule. (Right) Standard u-µP *with* our $1/\sqrt{\text{fan-out}}$ input LR scaling rule.

*Table 4.* Table of unit scaling factors for ops required for Llama, building on Table A.2. from the Unit Scaling paper (Blake et al., 2023).

| Op | Unit Scaling factors |
|---|---|
| $\text{attention}(q, k, v) =$ | $\alpha = \beta_q = \beta_k = \beta_v =$ |
| $\quad \text{softmax}\left(\alpha_{\text{attn}}\, d_{\text{head}}^{-1}\, (qk^\top)\, c_{\text{mask}}\right) v$ | $\quad 1/\log\_\text{interpolate}\left(\frac{1}{1+\frac{4d_{\text{head}}}{\alpha_{\text{attn}}^2}}, 1, \sqrt{\frac{\log(s)}{s}}\right)$ |
| $\text{gated\_silu}(x_{\text{in}}, x_{\text{gate}}) =$ | $\alpha = \beta_{x_{\text{in}}} = \beta_{x_{\text{gate}}} =$ |
| $\quad x_{\text{in}} \odot x_{\text{gate}} \odot \text{sigmoid}(\alpha_{\text{ffn-act}}\, x_{\text{gate}})$ | $\quad 1/\log\_\text{interpolate}\left(\frac{1}{1+\frac{1}{\alpha_{\text{ffn-act}}^2}}, \frac{1}{\sqrt{2}}, \frac{1}{2}\right)$ |
| $\text{residual\_add}(x_{\text{resid.}}, x_{\text{skip}}) = a\, x_{\text{resid.}} + b\, x_{\text{skip}}$ | $a = \sqrt{\frac{\tau}{\tau+1}},\ b = \sqrt{\frac{1}{\tau+1}}$ (see D.1.2 for full details, inc. values for $\tau$.) |
| $\text{RoPE}(x)$ | $\alpha = \beta = 1$ (i.e. no scaling) |
| $\text{RMSNorm}(x)$ (non-trainable, see (Lingle, 2024)) | $\alpha = \beta = 1$ (i.e. no scaling) |

# C. Unit-scaled pre-norm residual layers

The popular pre-norm residual network architecture is simple to implement, but problematic to combine with Unit Scaling. It exhibits scale-growth in the skip-stream at initialization, due to the repeated addition of residual connections without subsequent normalization. Here we present a surprising and useful finding: that for any pre-norm model there exists a mathematically-equivalent model where this scale-growth is eliminated, through the careful re-scaling of residual connections. This is a crucial preliminary step for our u-µP HP scheme, which is discussed in Appendix D (readers only interested in the final result may skip ahead to D.1.2).

## C.1. Scale growth in pre-norm residual networks

Let's consider a pre-norm residual network of depth $L$:

$$R_0(x) = r_0 x, \tag{1}$$

$$R_l(x) = r_l f_l(R_{l-1}(x)) + R_{l-1}(x), \quad l = 1, .., L \tag{2}$$

$$R_{L+1}(x) = f_{L+1}(R_L(x)) \tag{3}$$

with embedding multiplier $r_0$ and residual branch multipliers $r_l$ for $l = 1, .., L$. To satisfy pre-norm, all $f_l$ are zero-homogeneous functions, i.e. $f_l(\lambda x) = f_l(x)$.

The scale of the skip-stream at initialization as a result of Equation (2) is

$$\sigma(R_l) = \sqrt{r_l^2 \sigma(f_l)^2 + \sigma(R_{l-1})^2} > \sigma(R_{l-1}), \quad l = 1, .., L \tag{4}$$

assuming $r_l^2 \sigma(f_l)^2 > 0$. This shows that scale inevitably grows with the addition of each residual layer.

This scale-growth is clearly incompatible with unit scaling, which aims for $\sigma(R_l) = 1$ for $l = 0, .., L + 1$. In the following we present an elegant solution to this problem making use of a symmetry transformation available in pre-norm residual architectures.

## C.2. Residual symmetry in pre-norm architectures

To resolve the problem of scale shift in residual networks demonstrated by eq. (4), we try a slightly more general ansatz:

$$\hat{R}_0(x) = x, \tag{5}$$

$$\hat{R}_l(x) = a_l f_l(\hat{R}_{l-1}(x)) + b_l \hat{R}_{l-1}(x), \tag{6}$$

$$\hat{R}_{L+1}(x) = f_{L+1}(\hat{R}_L(x)) \tag{7}$$

with coefficients $a_l, b_l$. We want to choose these coefficients so that the outputs of $\hat{R}_l$ are unit-scaled if the outputs $f_l, \hat{R}_{l-1}$ are. A similar calculation as in eq. (4) leads to the sufficient condition

$$a_l^2 + b_l^2 = 1,$$

which can be easily satisfied. Having restored Unit Scale, we are faced with another issue. It seems that Equations (5) to (7) describes a different network than Equations (1) to (3), whereas ideally the relation from input to final output should be unchanged when converting the network to Unit Scaling.

Note that the coefficients $a_l, b_l$ are not uniquely defined yet, so our mathematical intuition tells us that we should find an additional constraint to get a unique solution. To find this constraint, let us consider our original residual network in Equations (1) to (3) and analyze how the variance propagates through the network if we assume all operations $f_l$ satisfy Unit Scaling and $\sigma(x) = 1$. Let $\sigma_{l-1}^2$ denote the variance of $R_{l-1}$. Then a simple inductive calculation shows that

$$\sigma_{l-1}^2 = \sum_{i=0}^{l-1} r_i^2.$$

By Equation (2) the output of $R_l$ adds a quantity of variance $r_l^2$ from the residual connection and a quantity of variance $\sigma_{l-1}^2$ from the skip connection. Intuitively, the ratio of these variances should be more important for the overall network dynamics than the absolute scales. Thus our constraint becomes preserving the ratio of variances from the original model, through our choice of $a_l, b_l$:

$$\frac{a_l^2}{b_l^2} = \frac{\sigma(r_l f_l)^2}{\sigma_{l-1}^2} = \frac{r_l^2}{\sum_{i=0}^{l-1} r_i^2} =: \tau_l,$$

which (up to sign) uniquely defines our multipliers $a_l, b_l$ as

$$a_l = \sqrt{\frac{\tau_l}{\tau_l + 1}}, \quad b_l = \sqrt{\frac{1}{\tau_l + 1}} \tag{8}$$

In summary, we propose the modified residual network

$$\hat{R}_0(x) = x, \tag{9}$$

$$\hat{R}_l(x) = \sqrt{\frac{\tau_l}{\tau_l + 1}} f_l(\hat{R}_{l-1}(x)) + \sqrt{\frac{1}{\tau_l + 1}} \hat{R}_{l-1}(x), \tag{10}$$

$$\hat{R}_{L+1}(x) = f_{L+1}(\hat{R}_L(x)), \tag{11}$$

$$\tau_l = \frac{r_l^2}{\sum_{i=0}^{l-1} r_i^2}. \tag{12}$$

Our main result of this section is that this transformation is indeed a mathematical equivalence under a simple additional structural assumption:

*Lemma* C.1. Consider $R_l$, $\hat{R}_l$ defined as in Equations (2) and (10). Then $\hat{R}_l = R_l / \sqrt{\sum_{i=0}^{l} r_i^2}$ for all $l = 0, .., L$.

Remarkably, this result does not assume the individual network operations $f_l$ to actually satisfy Unit Scaling. It is purely a consequence of the pre-norm residual structure. However, only under Unit Scaling can the factors $\tau_l$ be interpreted as the ratio of variances between skip and residual branch.

As a consequence of the lemma, the final residual output $R_L(x)$ is the same as in our original network up to a fixed multiplier. Due to the zero-homogeneity of the final output function $f_{L+1}$ this gives $\hat{R}_{L+1} = f_{L+1}\left( R_L(x) / \sqrt{\sum_{i=0}^{l} r_i^2} \right) = f_{L+1}(R_L(x)) = R_{L+1}$, proving the mathematical equivalence of our residual scheme.

Modern LLM architectures like Llama (Touvron et al., 2023) are pre-norm residual networks of this kind. Hence they admit a faithful unit-scaled reparametrization.

### C.3. Unit Scaling for transformer residuals

The above scheme describes Unit Scaling for arbitrary pre-norm residual networks. We now apply it to the case of (pre-norm) transformer residual layers.

We can describe a transformer in terms of the residual network given in Equations (1) to (3). Our $f_l$ functions alternate between self-attention layers and feed-forward layers. Implementations differ in the handling of how residual multipliers $r_l$ correspond to HPs. In many cases practitioners simply ignore these $r_l$, but for the sake of expressivity we assume the two types of residual layer each have their own HP, as well as the embedding. In other words,

$$
r_l = \begin{cases} \alpha_{\text{emb}} & l = 0 \\ \alpha_{\text{attn-residual}} & l \text{ is odd} \\ \alpha_{\text{ffn-residual}} & l \text{ is even, and } l > 0. \end{cases}
$$

To convert this to a Unit Scaled network we apply Equations (9) to (12), from which can derive the following closed-form expression for $\tau_l$:

$$
\tau_l = \begin{cases} \dfrac{\alpha_{\text{attn-residual}}}{\alpha_{\text{emb}} + \ell\alpha_{\text{attn-residual}} + \ell\alpha_{\text{ffn-residual}}} & l \text{ is odd} \\ \dfrac{\alpha_{\text{ffn-residual}}}{\alpha_{\text{emb}} + (\ell+1)\alpha_{\text{attn-residual}} + \ell\alpha_{\text{ffn-residual}}} & l \text{ is even}. \end{cases}
$$

where $\ell = \lfloor \frac{l-1}{2} \rfloor$.

### C.4. Proof of Lemma C.1

*Proof.* This is proved by induction. For the base-case $l = 1$, we have $\tau_1 = r_1^2/r_0^2$, giving

$$
\begin{aligned}
\hat{R}_1(x) &= \sqrt{\frac{\tau_l}{\tau_l + 1}} f_1(x) + \sqrt{\frac{1}{\tau_l + 1}} x \\
&= (r_1 f_1(x) + r_0 x) / \sqrt{r_0^2 + r_1^2} \\
&= R_1 / \sqrt{r_0^2 + r_1^2}.
\end{aligned}
$$

Then if the statement holds for $l - 1$ we have

$$\hat{R}_l(x) = \sqrt{\frac{\tau_l}{\tau_l + 1}} f_l(\hat{R}_{l-1}(x)) + \sqrt{\frac{1}{\tau_l + 1}} \hat{R}_{l-1}(x)$$

$$= \frac{r_l}{\sqrt{\sum_{i=0}^{l} r_i^2}} f_l(\hat{R}_{l-1}(x)) + \frac{\sqrt{\sum_{i=0}^{l-1} r_i^2}}{\sqrt{\sum_{i=0}^{l} r_i^2}} \hat{R}_{l-1}(x)$$

$$= \left( r_l f_l(\hat{R}_{l-1}(x)) + \sqrt{\sum_{i=0}^{l-1} r_i^2} \hat{R}_{l-1}(x) \right) / \sqrt{\sum_{i=0}^{l} r_i^2}$$

$$= \left( r_l f_l(R_{l-1}(x)) + \sqrt{\sum_{i=0}^{l-1} r_i^2} \frac{R_{l-1}(x)}{\sqrt{\sum_{i=0}^{l-1} r_i^2}} \right) / \sqrt{\sum_{i=0}^{l} r_i^2}$$

$$= \left( r_l f_l(R_{l-1}(x)) + R_{l-1}(x) \right) / \sqrt{\sum_{i=0}^{l} r_i^2}$$

$$= R_l(x) / \sqrt{\sum_{i=0}^{l} r_i^2}$$

$\square$

## D. Hyperparameters for u-µP

Here we explain the HPs from Section 3.3 in more detail. Our desiderata for the u-µP HPs are as follows:

1. **Unit scale:** For every choice of HPs we satisfy Unit Scaling, meaning that the variance at initialization throughout the entire model is 1.

2. **Interpretable HPs:** Each HP value determines a dynamic of the model at initialization that we consider important, giving them a clear interpretation.

3. **Fully expressive:** They result in a model which is as expressive as a standard pre-norm transformer network, meaning that for any model expressed by Equations (1) to (3), there is a choice of HPs that forms a mathematically-equivalent u-µP residual model.

4. **No redundancy:** Removing any HP results in a strictly less expressive model.

### D.1. Residual branch multipliers $\alpha_{\text{residual}}$, $\alpha_{\text{residual-attn-ratio}}$

In this section we present our u-µP residual branch multipliers. They can be viewed as a reparametrization of the original residual multipliers in C.3. We begin by explaining our heuristic for our new set of residual HPs and then combine this with the residual branch re-scaling derived in the previous section, which gives our u-µP residual scheme.

#### D.1.1. IMPROVED HPs FOR TRANSFORMER RESIDUALS

In Section 3.3 we refer to a new pair of u-µP HPs, $\alpha_{\text{residual}}$ and $\alpha_{\text{residual-attn-ratio}}$, which we use for residual layers. Here we define them and make the case for including them as part of our u-µP scheme. To avoid cluttered notation, in this section we rename

$$\alpha_{\text{residual}} = \alpha_r, \quad \alpha_{\text{residual-attn-ratio}} = \alpha_\rho.$$

There is no clear convention for the set of multipliers in a standard residual model. Hence we adopt the most generous and straightforward group of multipliers, $(\alpha_{\text{emb}}, \alpha_{\text{attn-residual}}, \alpha_{\text{ffn-residual}})$, as in Section C.3. For simplicity we rename

$$\alpha_{\text{emb}} = \alpha_e, \quad \alpha_{\text{attn-residual}} = \alpha_a \quad \alpha_{\text{ffn-residual}} = \alpha_f.$$

To make the presentation more clear, we derive our new HPs using the standard residual scheme from Equations (1) to (3). For the actual unit scaled implementation one needs to transform the multipliers following Equations (9) to (12), which we do in Section D.1.2.

Our new multipliers satisfy the following properties that $(\alpha_e, \alpha_a, \alpha_f)$ do not:

1. They have an intuitive interpretation for the multiplier values in the context of the residual output $R_L(x)$, such that each controls a dynamic in the model that we consider important.

2. The number of multipliers is minimized, under the constraint that expressivity is maintained.

3. The most effective choice of one multiplier depends as little as possible on the choice of the other multiplier(s).

To facilitate our analysis, we can view the transformer residual output as the sum of three terms:

$$R_L = R_L^{(e)} + R_L^{(a)} + R_L^{(f)},$$
$$R_L^{(e)} := \alpha_e x,$$
$$R_L^{(a)} := \sum_{l=1}^{L/2} \frac{\alpha_a}{\sqrt{L}} f_{2l-1}(R_{2l-1}(x)),$$
$$R_L^{(f)} := \sum_{l=1}^{L/2} \frac{\alpha_f}{\sqrt{L}} f_{2l}(R_{2l}(x)),$$
$$R_L^{(r)} := R_L^{(a)} + R_L^{(f)},$$

Note that we have added in the depth-µP multipliers here, though a similar analysis can be performed for non-depth-µP models. As above, $f_l$ functions alternate between self-attention layers and feed-forward layers.

With respect to point 1., we propose two new multipliers that correspond to dynamics in the network which we suggest are important to control at initialization. The first is the ratio of the scale of the residuals' contributions to those of the embedding, $\alpha_r = \sigma(R_L^{(r)})/\sigma(R_L^{(e)})$. The second is the ratio of the scale of the attention-residuals' contributions to those of the feed-forward-residuals, $\alpha_\rho = \sigma(R_L^{(a)})/\sigma(R_L^{(f)})$. We now demonstrate how the existing $(\alpha_e, \alpha_a, \alpha_f)$ multipliers can be replaced by $(\alpha_r, \alpha_\rho)$.

Let us first examine these two quantities under the standard set of HPs. Residual functions of the same kind have the same expected output scale at initialization in pre-norm networks. Hence we denote the output scale $\sigma(f_l(R_l))$ of self-attention functions as $\sigma_a$, and of feed-forward functions as $\sigma_f$. We thus have the following scales at the output:

$$\sigma(R_L^{(e)}) = \alpha_e \sigma(x),$$
$$\sigma(R_L^{(a)}) = \frac{\alpha_a}{\sqrt{L}} \sigma \left( \sum_{i=1}^{L/2} f_{2l-1}(R_{2l-1}) \right) = \frac{\alpha_a \sigma_a}{\sqrt{2}},$$
$$\sigma(R_L^{(f)}) = \frac{\alpha_f}{\sqrt{L}} \sigma \left( \sum_{i=1}^{L/2} f_{2l}(R_{2l}) \right) = \frac{\alpha_f \sigma_f}{\sqrt{2}},$$
$$\sigma(R_L^{(r)}) = \sqrt{\sigma(R_L^{(a)})^2 + \sigma(R_L^{(f)})^2} = \frac{\sqrt{(\alpha_a \sigma_a)^2 + (\alpha_f \sigma_f)^2}}{\sqrt{2}},$$

meaning our new multipliers are equal to:

$$\alpha_\rho = \frac{\alpha_a}{\alpha_f}\frac{\sigma_a}{\sigma_f},$$

$$\alpha_r = \frac{\sqrt{(\alpha_a\sigma_a)^2 + (\alpha_f\sigma_f)^2}}{\sqrt{2}\,\alpha_e\sigma(x)},$$

$$= \sqrt{\frac{\alpha_\rho^2 + 1}{2}}\,\frac{\sigma_f}{\sigma(x)}\frac{\alpha_f}{\alpha_e}.$$

The original $\alpha_a, \alpha_f$ multipliers can then be written in terms of $\alpha_r, \alpha_\rho$:

$$\alpha_a = \alpha_\rho\alpha_f\frac{\sigma_f}{\sigma(\sigma_a)}$$

$$\alpha_f = \alpha_r\alpha_e\frac{\sigma(x)}{\sigma_f}\sqrt{\frac{2}{\alpha_\rho^2 + 1}}$$

We have replaced two of the three original multipliers, but still have a dependence on $\alpha_e$ here in our expressions for $\alpha_f$ and $R_L^{(e)}$, which we now remove by dividing it out of our residual branches and embedding. We use the hat ($\hat{\cdot}$) symbol to denote terms that have been divided-through by $\alpha_e$. This new system of equations is equivalent to our old one thanks to the zero-homogeneity of the final post-residual layer:

$$R_{L+1}(x) = f_{L+1}(R_L^{(e)} + R_L^{(r)})$$
$$= f_{L+1}((R_L^{(e)} + R_L^{(r)})/\alpha_e)$$
$$= f_{L+1}(\hat{R}_L^{(e)} + \hat{R}_L^{(r)})$$

This gives $\hat{R}_L^{(e)} = \alpha_e x/\alpha_e = x$, removing our first occurrence of $\alpha_e$. Following the division through $\hat{R}_L^{(r)}$ results in:

$$\hat{R}_L^{(r)} = \hat{R}_L^{(a)} + \hat{R}_L^{(f)},$$

$$\hat{R}_L^{(a)} := \sum_{l=1}^{L/2}\frac{\hat{\alpha}_a}{\sqrt{L}}f_{2l-1}(R_{2l-1}),$$

$$\hat{R}_L^{(f)} := \sum_{l=1}^{L/2}\frac{\hat{\alpha}_f}{\sqrt{L}}f_{2l}(R_{2l}),$$

$$\hat{\alpha}_a = \alpha_\rho\hat{\alpha}_f\frac{\sigma_f}{\sigma_a},$$

$$\hat{\alpha}_f = \alpha_r\frac{\sigma(x)}{\sigma_f}\sqrt{\frac{2}{\alpha_\rho^2 + 1}}.$$

This system of equations is the same as the original, but with the two $\alpha_e$ terms dropped, meaning our model's multipliers can be expressed in terms of only $\alpha_r$ and $\alpha_\rho$. Using the above equations, any pair of values for $(\alpha_r, \alpha_\rho)$ can be translated back into an equivalent set of values for $(\alpha_e, \alpha_a, \alpha_f)$ such that the output $R_{L+1}(x)$ is the same, meaning that our multipliers are no less expressive than the original set. This satisfies our desired property of minimizing the number of multipliers while maintaining expressivity.

We can simplify further in the case of unit-scaled models, which are designed such that $\sigma(x), \sigma_a, \sigma_f$ are all 1 at initialization.

In this case our re-parametrization becomes:

$$\hat{\alpha}_a = \alpha_\rho \hat{\alpha}_f, \tag{13}$$

$$\hat{\alpha}_f = \alpha_r \sqrt{\frac{2}{\alpha_\rho^2 + 1}}, \tag{14}$$

$$\hat{\alpha}_e = 1. \tag{15}$$

This is the basis of our claim that Unit Scaling enables this more intuitive set of multipliers. Not only do the multipliers $\alpha_r$ and $\alpha_\rho$ represent important dynamics in the network at initialization (the ratio of residual-to-embedding scales, and the ratio of attention-to-feed-forward scales), but it's only via unit scaling that these equations become simple enough to implement in practice. Using equations Equations (13) to (15) for a non-unit scaled network may still be effective, but the interpretation we've given to $\alpha_r$ and $\alpha_\rho$ no longer hold.

Our final desired property is an empirical one: that the most effective choice of one multiplier depends as little as possible on the choice of the other multiplier(s). We demonstrate that our multipliers satisfy this property better than the standard set of residual multipliers (see Fig 11 and Fig 12).

### D.1.2. THE FULL U-µP RESIDUAL SCHEME

Here we give the full definition of our u-µP residual scheme, summarizing the results of previous sections. A general pre-norm transformer is implemented as:

$$R_0(x) = c\, x, \tag{16}$$
$$R_l(x) = a_l f_l(R_{l-1}(x)) + b_l R_{l-1}(x), \quad l = 1, .., L \tag{17}$$
$$R_{L+1}(x) = f_{L+1}(R_L(x)), \tag{18}$$

where $a_l, b_l$ and $c$ are scalar multipliers, and the $f_l$ alternate between self-attention and feed-forward layers. We consider our baseline set of residual HPs here to be $(\alpha_{\text{emb}}, \alpha_{\text{attn-residual}}, \alpha_{\text{ffn-residual}})$, which we implement (assuming depth-µP branch scaling) as:

$$a_l = \begin{cases} \dfrac{\alpha_{\text{attn-residual}}}{\sqrt{L}} & l \text{ is odd (self-attention)} \\[2ex] \dfrac{\alpha_{\text{ffn-residual}}}{\sqrt{L}} & l \text{ is even (feed-forward)} \end{cases}$$

$$b_l = 1$$

$$c = \alpha_{\text{emb}}.$$

The corresponding u-µP set of residual HPs is $(\alpha_{\text{residual}}, \alpha_{\text{residual-attn-ratio}})$, which we implement as:

$$a_l = \sqrt{\frac{\tau_l}{\tau_l + 1}} \tag{19}$$

$$b_l = \sqrt{\frac{1}{\tau_l + 1}} \tag{20}$$

$$c = 1, \tag{21}$$

$$\tau_l = \frac{1}{\sqrt{L}} \cdot \begin{cases} \dfrac{\hat{\alpha}_a}{1 + \ell\hat{\alpha}_a + \ell\hat{\alpha}_f} & l \text{ is odd} \\[2ex] \dfrac{\hat{\alpha}_f}{1 + (\ell+1)\hat{\alpha}_a + \ell\hat{\alpha}_f} & l \text{ is even} \end{cases}, \quad \ell = \left\lfloor \frac{l-1}{2} \right\rfloor \tag{22}$$

$$\hat{\alpha}_a = \hat{\alpha}_f\, \alpha_{\text{residual-attn-ratio}} \tag{23}$$

$$\hat{\alpha}_f = \sqrt{\frac{2}{\alpha_{\text{residual-attn-ratio}}^2 + 1}}\, \alpha_{\text{residual}}. \tag{24}$$

This is the u-μP residual scheme. It satisfies the three properties that we initially set out to achieve: the variance at initialization of our $R_l(x)$ is always 1, our HPs have a clear and useful interpretation, and our scheme is as expressive as the baseline (which is neither unit-scaled or has interpretable HPs).

### D.2. Multipliers for non-homogeneous ops $\alpha_{\text{attn-softmax}}$, $\alpha_{\text{ffn-act}}$, $\alpha_{\text{loss-softmax}}$

In this section we derive the rest of our u-μP multipliers. We want to identify the minimal set of multipliers that can still express all different choices of pre-op scales in the model. The crucial observation is that every pre-scale multiplier $\alpha$ of an operation $h \mapsto f(\alpha h)$ can be propagated through the network if $f$ is $k$-homogeneous for some $k > 0$, i.e. $f(\alpha x) = \alpha^k f(x)$. We can iterate this along the computational path until either the next operation is non-homogeneous, we are at the end of a residual path, or the next operation is 0-homogeneous (e.g. a norm). In the first case the accumulated scales are absorbed in the pre-op scale of the non-homogeneous operation (where we introduce a multiplier), in the second case they are absorbed in the residual addition for that branch (where we again introduce a multiplier), and in the final case the scale disappears (so we start over). We now go through the Llama forward computation and follow this paradigm to identify our multipliers in Table 5.

*Table 5.* A walkthrough of the Llama architecture, showing how our $\alpha_{\text{attn-softmax}}$, $\alpha_{\text{ffn-act}}$ and $\alpha_{\text{loss-softmax}}$ multipliers are derived via an analysis of scale-propagation.

| Op | Scale propagation behavior |
| --- | --- |
| Embedding | We already saw in the previous section that the embedding multiplier can be absorbed in the residual multipliers. |
| Attention RMSNorm | This operation is 0-homogeneous and thus we start over. |
| Query and key | Query and key itself are linear, hence their weight multipliers get propagated. |
| Query-Key matmul | The query-key matrix multiplication is 2-homogeneous when viewed as function of the concatenated query-key vector. Hence it propagates the scale. |
| Softmax | The softmax operation is non-homogeneous. Thus the pre-op scale of the softmax becomes our first multiplier $\alpha_{\text{attn-softmax}}$. |
| Value | The value layer is linear and hence propagates its scale. |
| Softmax-value matmul | This operation is linear in all arguments and hence propagates the scale. |
| Attention projection | This operation is linear and lies at the end of the attention residual path. Hence there are no more multipliers in the attention block. |
| FFN RMSNorm | This operation is 0-homogeneous and thus we start over. |
| FFN input scale | The input layer is linear, hence its weight multiplier gets propagated. |
| Sigmoid input | This function is non-homogeneous and thus we have another multiplier $\alpha_{\text{ffn-act}}$. |
| SiLU weight | This layer is also linear and propagates the scale. |
| Product | The entry-wise multiplication of the outputs of sigmoid, input layer and SiLU weight is homogeneous and thus propagates the scale. |
| FFN output | This layer is linear and at the end of the residual path. Hence there are no more multipliers in the FFN residual block. |
| Output RMSNorm | This operation is 0-homogeneous and thus we start over. |
| Output head | This layer is linear, hence its weight multiplier gets propagated. |
| Loss | The cross-entropy loss is non-homogeneous and leads to our final multiplier $\alpha_{\text{loss-softmax}}$. |

In summary, we have three multipliers $\alpha_{\text{attn-softmax}}$, $\alpha_{\text{ffn-act}}$, $\alpha_{\text{loss-softmax}}$ that are applied in the softmax, sigmoid and

| | |
|---|---|
| Dataset | WikiText-103 (Merity et al., 2017) |
| Sequence Length | 256 |
| Vocab Size | 32000 |
| Training Set Tokens | 138 M |
| Architecture | Llama (Touvron et al., 2023) (Transformer, PreNorm, RMSNorm, SwiGLU, RoPE, "untied" embeddings), non-trainable RMSNorm parameters. |
| Width | 256 |
| Depth | 4 |
| Head Dimension | 64 |
| Batch size | 64 |
| Training steps | 8192 (0.97 epochs) |
| LR schedule | Cosine to $10\%$, 2000 steps warm-up |
| Optimizer | AdamW $(\beta_1, \beta_2, \epsilon) = (0.9, 0.999, 10^{-8})$ |
| Weight Decay | $2^{-13}$, independent (Loshchilov & Hutter, 2019) |
| Dropout | 0.0 |
| µP HP Search Range | $\eta \in [2^{-10}, 2^{-6}]$ $\hat{\eta}_{\text{emb}} \in [2^0, 2^8]$ $\sigma_{\text{init}}, \alpha_{\text{emb}}, \alpha_{\text{attn}}, \alpha_{\text{output}} \in [2^{-2}, 2^2]$ |
| u-µP HP Search Range | $\eta \in [2^{-1}, 2^3]$ $\alpha_{\text{attn}} \in [2^{-2}, 2^2]$ $\alpha_{\text{residual}}, \alpha_{\text{residual-attn-ratio}}, \alpha_{\text{ffn-act}}, \alpha_{\text{output}} \in [2^{-3}, 2^3]$ |
| µP HP Defaults | $\sigma_{\text{init}} = \alpha_{\text{emb}} = \alpha_{\text{attn}} = \alpha_{\text{output}} = \hat{\eta}_{\text{emb}} = 1$ |
| u-µP HP Defaults | $\alpha_{\text{residual}} = \alpha_{\text{residual-attn-ratio}} = \alpha_{\text{ffn-act}} = \alpha_{\text{output}} = \alpha_{\text{attn}} = 1$ |

*Table 6.* Default hyperparameters and training settings.

loss function via:

$$f_{\text{softmax}}(q, k) = \text{softmax}(\alpha_{\text{attn-softmax}} \cdot d_{\text{head}}^{-1} \cdot (q \cdot k^t)), \tag{25}$$

$$f_{\text{act}}(h) = \text{sigmoid}(\alpha_{\text{ffn-act}} \cdot h), \tag{26}$$

$$f_{\text{loss-softmax}}(h, x_{\text{targets}}) = \text{CE}(\alpha_{\text{loss-softmax}} \cdot h, x_{\text{targets}}) \tag{27}$$

We do not explicitly show the derivation of the residual multipliers here, as they undergo a change in accordance with D.1.1 before we get our final $\alpha_{\text{residual}}$ and $\alpha_{\text{residual-attn-ratio}}$.

Our analysis from above shows that these three multipliers together with the residual multipliers are as expressive as the full set of pre-ops multipliers in the whole transformer architecture while having no redundancy, i.e. a change in one of the multipliers cannot equivalently be expressed in terms of changes to the other multipliers.

## E. Additional Experimental Details

To compare µP and u-µP with the Llama architecture on a larger dataset, we modify the implementation provided by Yang et al. (2022) for µP and implement u-µP in the same framework.

### E.1. Hyperparameter Independence

Our analysis indicates that µP's hyperparameters have overlapping effects on dynamics-defining scales within the model, while u-µP attempts to isolate their effect. We hypothesize that this effect should be visible in the final loss—the effects of

u-µP's hyperparameters should be more separable than that of µP's.

In our first test of this hypothesis, we construct pairs of hyperparameters, and perform a coarse 2D sweep for each pair (Figure 11, Figure 12). These results show some visual dependence between µP hyperparameters as a diagonal structure in the grids, such as $(\hat{\eta}_{\text{emb}}, \sigma_{\text{init}})$ and $(\eta, \alpha_{\text{attn}})$. We quantify this difference by evaluating the increase in loss on a given row by using the argmin hyperparameter from a different row of the grid, compared with the actual minimum and averaged over all grids. This metric gives an average loss increase of 0.08 for µP versus 0.03 for u-µP. This suggests a quantifiable improvement in hyperparameter separability, but note that the metric may conflate this with the flatness of the optimum.

The second test is more directly practical. We compare two hyperparameter search methods on µP and u-µP. The first is a random grid search, which samples configurations without replacement from a grid defined over all hyperparameters. After performing a single search, we can simulate the effect of a shorter search by taking a random sample of the results. The second method is an independent search, which consists of the following phases:

1. Perform a 1D line search for an optimal learning rate, with other hyperparameters set to their default (9 runs).

2. For each hyperparameter in parallel, perform a 1D line search (330 runs).

3. Combine the best settings from step 2, and re-evaluate (6 runs).

Each 1D line search can be done on an iteratively refined grid, to provide an incremental improvement as the number of runs increases.

Our results from this test in Figure 3 show that the first LR sweep is much more efficient for u-µP since the default hyperparameters are better. For this reason, the 1D line search can outperform a random grid. We also observe that the final step of combining optimum hyperparameters is very harmful to µP, while it shows only a slight degradation for u-µP, which was expected as a regression to the mean.

### E.2. Hyperparameter Transfer

We compare learning rate transfer for µP and u-µP in Figures 1 and 5, over a logarithmic grid of spacing $2^{1/2}$, with 3 runs for each point. We observe:

1. u-µP transfer of LR over width, training steps, batch size and depth is similar to or better than µP, when starting from default parameters.

2. u-µP and µP both show increased variance when the learning rate is too high (visible in wide confidence intervals in Figure 5).

3. The default settings for u-µP are better than those of µP, especially when scaling width and training duration (steps or batch size).

Moreover, in Figures 2 and 6 we highlighted the poor transfer of the embedding LR multiplier $\hat{\eta}_{\text{emb}}$ in µP, and demonstrated that the $1/\sqrt{\text{fan-out}}$ scaling rule in u-µP resolves this issue. For completeness, we now include in Figure 7 a comparison of the width transfer of all model hyperparameters from both µP and u-µP. We find good transfer for all hyperparameters defined in the u-µP scheme, and a clear visualization of how setting all u-µP multiplier defaults to 1 (i.e., just omitting all of the available scale modifiers) results in near optimal performance. In contrast, the µP results indicate that optimum values for $\sigma_{\text{init}}$ and $\hat{\eta}_{\text{emb}}$ do not transfer over width in our real-world evaluation.

### E.3. Numerical Properties

Our analysis of the numerical properties of u-µP focuses on the RMS statistics of tensors that we wish to cast to FP8: linear module input activations, weights and output gradients. RMS captures the larger of the mean and scale of a distribution, and as such can be a good test of whether the tensor is likely to suffer range (clipping) errors in low-precision number formats.

Figure 4 shows the distribution of statistics over all linear modules in the model, and Figure 9 shows RMS on a per-tensor basis, as it evolves during training. From these, we note:

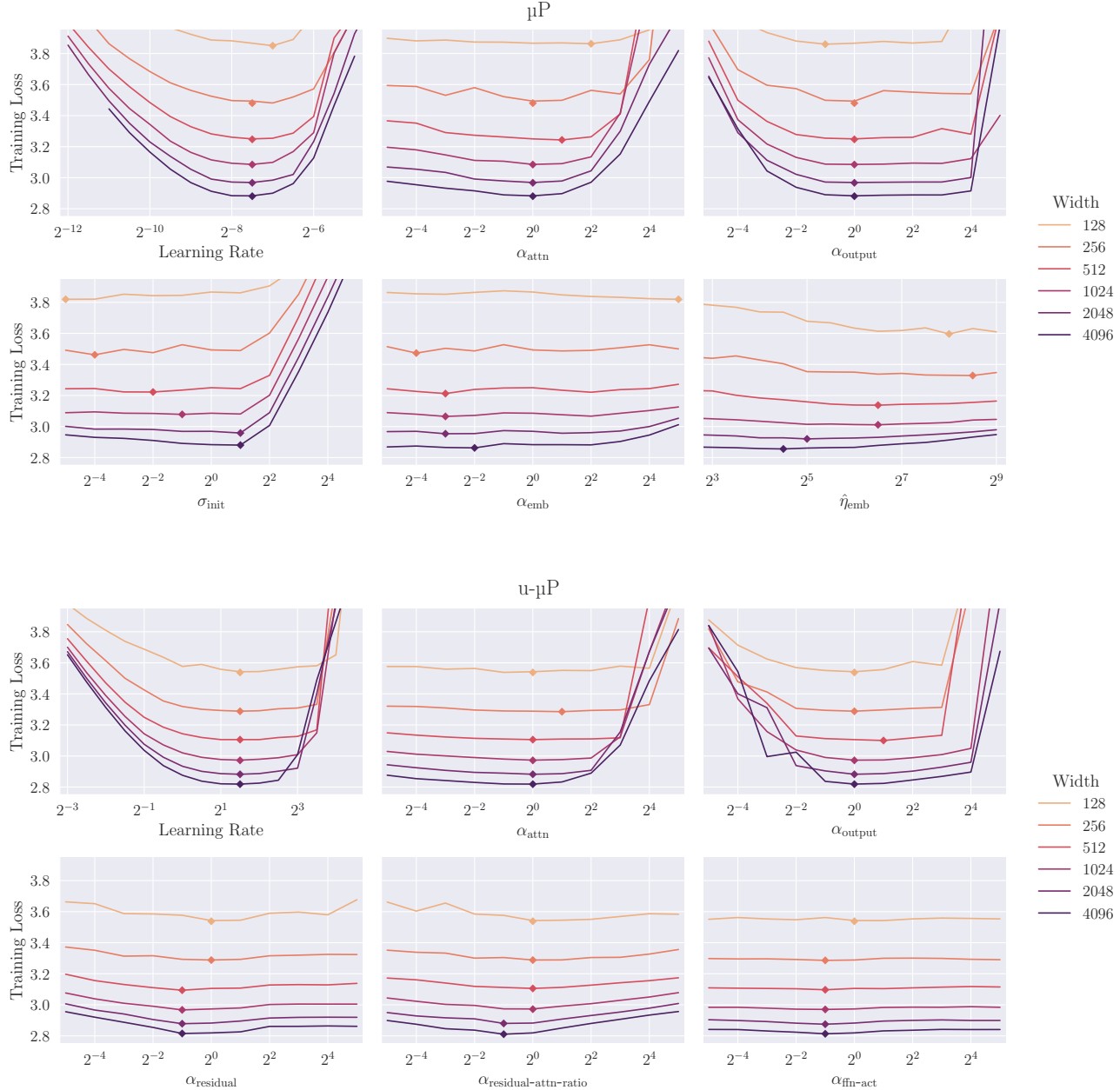

*Figure 7.* Transfer of model hyperparameters over width for µP (top) and u-µP (bottom). When one hyperparameter is being swept, all others are fixed at 1, with the exception of Learning Rate $\eta = (2^{1.5}, 2^{-7.5})$ for (u-µP, µP).

1. μP has gradients and weights with low RMS, at risk of FP8 underflow, whereas u-μP starts with $\mathrm{RMS} \approx 1$.

2. Many input activations do not grow RMS during training (due to a preceding non-trainable RMSNorm), however the attention out projection and FFN down projection have unconstrained input activations that grow considerably during training.

3. The decoder weight grows during training. Since it is preceded by a RMSNorm, the model may require scale growth in order to increase the scale of softmax inputs. Other weights grow slightly during training.

4. Gradients grow quickly but stabilize, except for attention out projection and FFN down projection, whose gradients shrink as the inputs grow.

We also evaluate how RMS growth is affected by model and training hyperparameters in the tensors that showed the highest end-training RMS, shown in Figure 10. This shows that the main parameter affecting scale growth is learning rate, with end-training RMS increasing to the right of the optimal LR basin, as training becomes unstable. End-training RMS is remarkably stable as width, depth, training steps and batch size are independently increased.

We therefore propose the FP8 scheme described in Section 5.1, which works for u-μP without any dynamic scaling or exponent bias search (see Figure 1 (top), Figure 8).

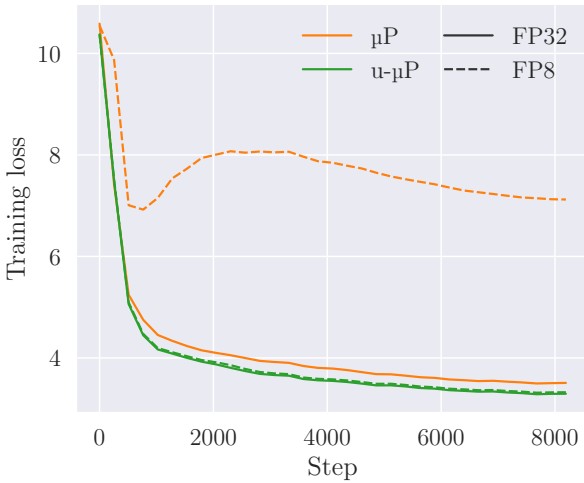

*Figure 8.* FP8 training by direct cast, width 256, default hyperparameters, $\eta = (2^1, 2^{-8})$ for (u-µP, µP).

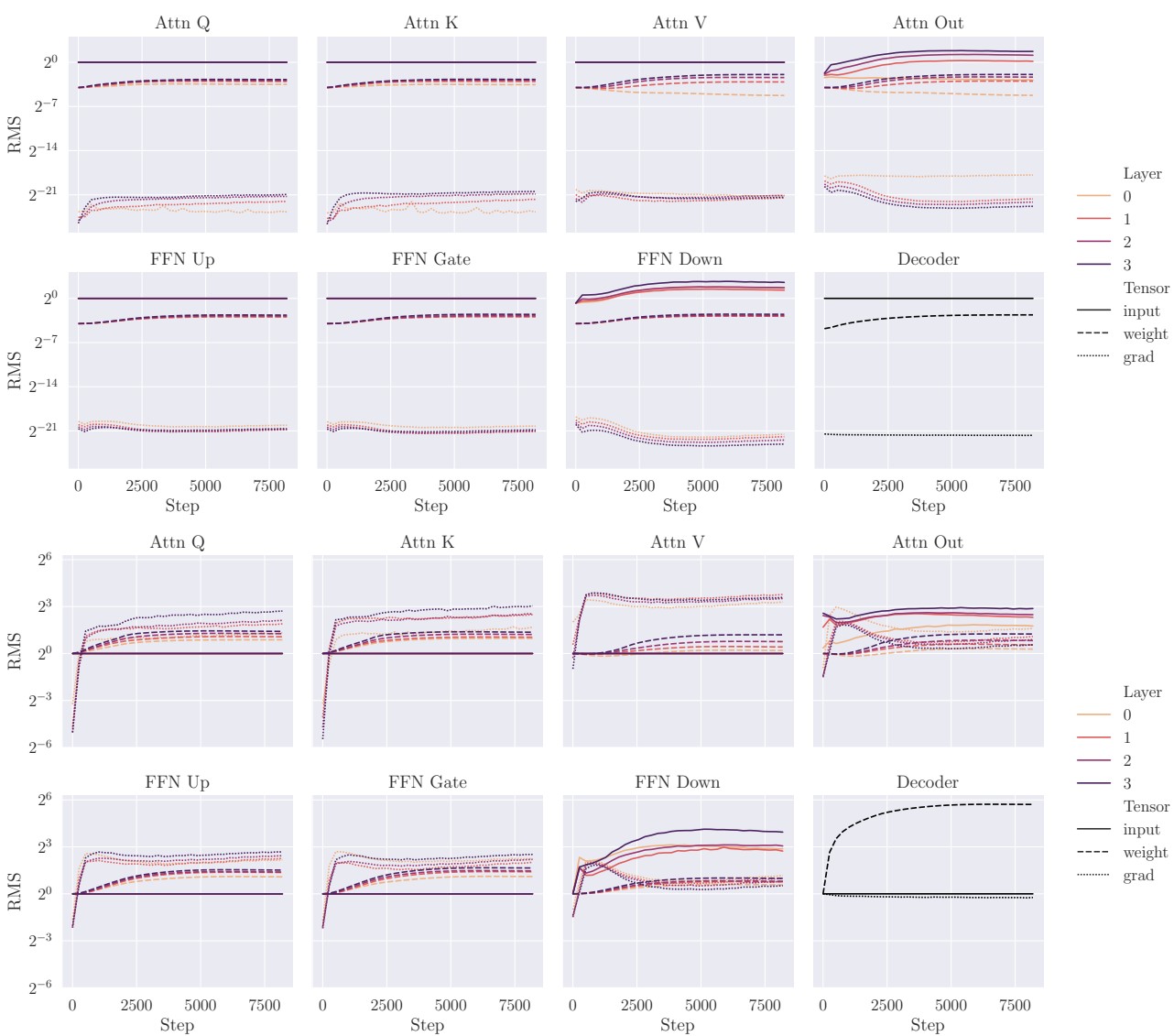

*Figure 9.* RMS during training, for all parametrized matmul inputs, for µP (top) and u-µP (bottom). Model width 256, default hyperparameters, $\eta = (2^1, 2^{-8})$ for (u-µP, µP).

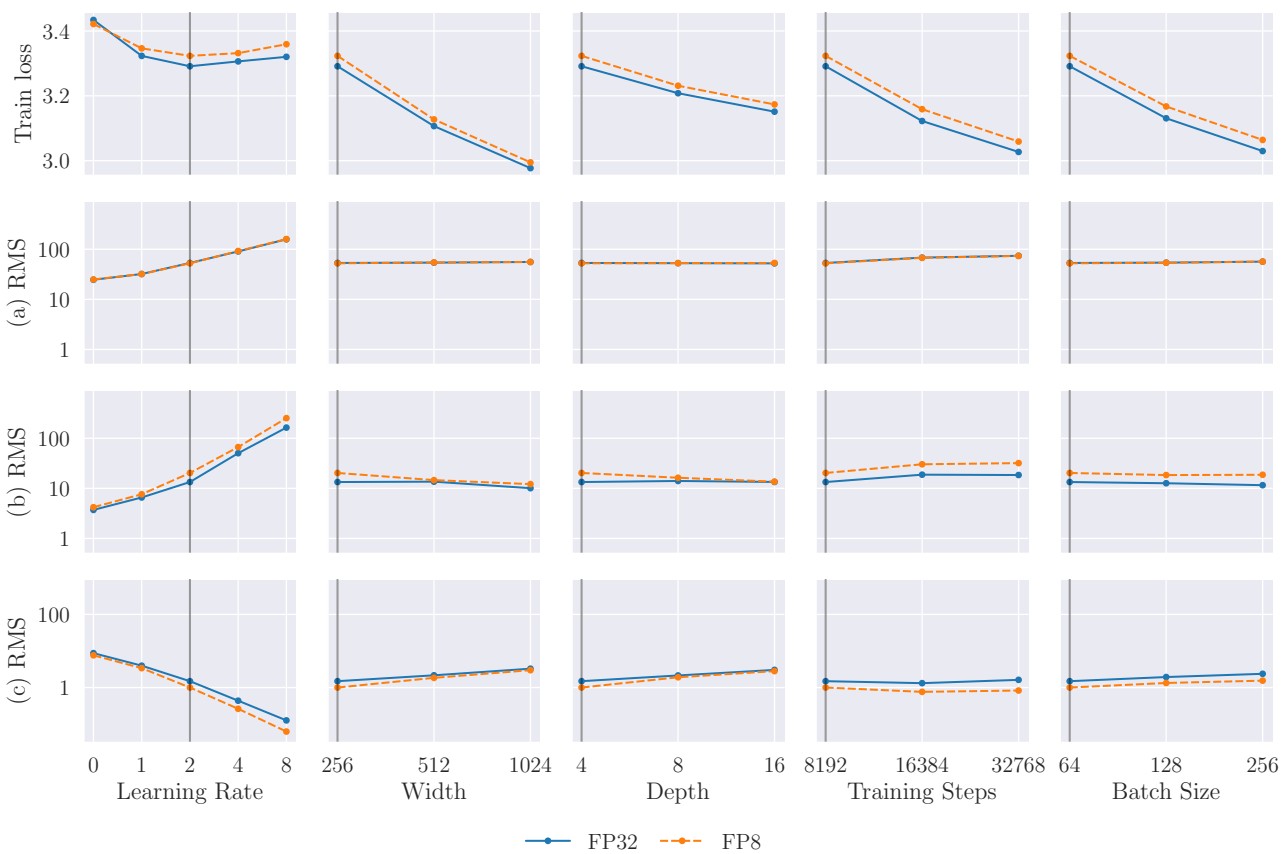

*Figure 10.* The effect of hyperparameters on FP8 training loss and on the end-training RMS of various tensors: (a) decoder weight, (b) last-layer FFN down-projection input and (c) last-layer FFN down-projection output gradient. Only learning rate has a substantial effect on the end-training RMS. Vertical lines show the default setting of that hyperparameter, as used for all other plots.

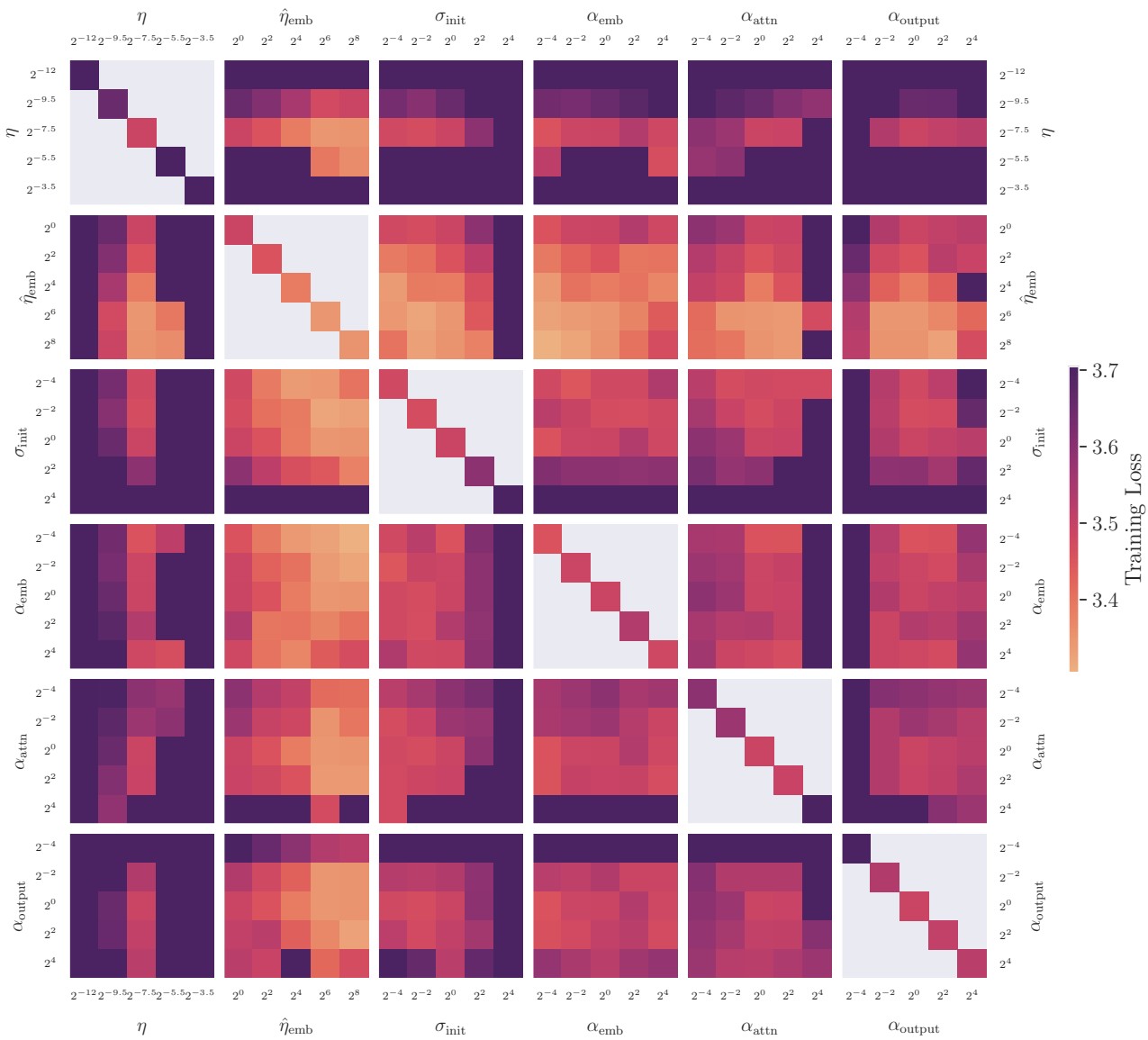

*Figure 11.* Hyperparameter coupling sweep for µP. Note strong coupling between optima, e.g. in the cases of $(\eta_{\mathrm{emb}}, \sigma_{\mathrm{init}})$ and $(\eta, \alpha_{\mathrm{attn}})$. See also: u-µP, Figure 12. Across all grids, the average training loss degradation from using the optimum from the wrong row/column is 0.08, which is worse than u-µP (0.03).

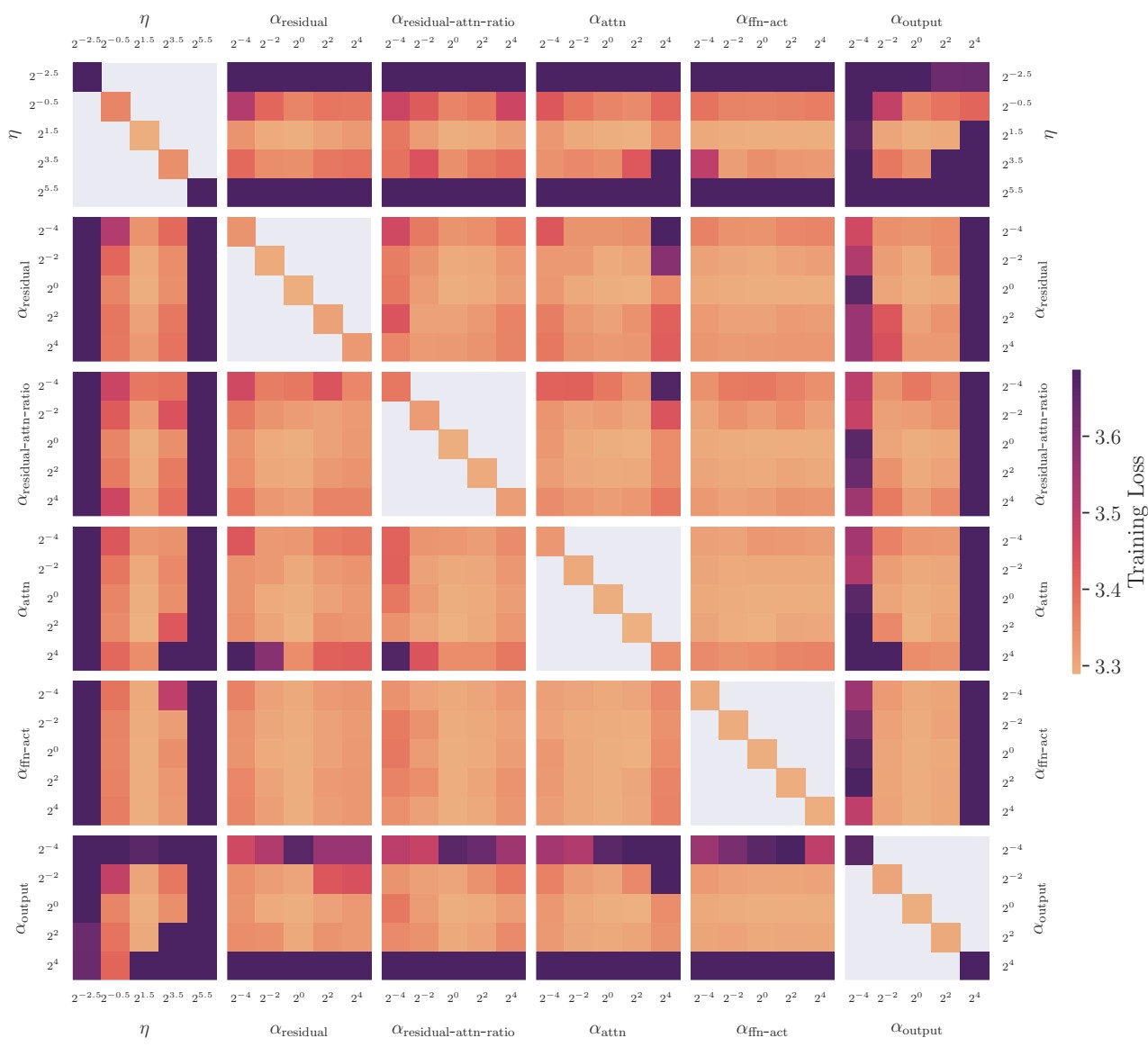

*Figure 12.* Hyperparameter coupling sweep for u-μP. Note less coupling than with μP, see Figure 11. Across all grids, the average training loss degradation from using the optimum from the wrong row/column is 0.03, which is better than μP (0.08).

