# OpenReview forum: "u-μP: The Unit-Scaled Maximal Update Parametrization"
_ICML.cc/2024/Workshop/WANT — WANT@ICML 2024 Poster_

### Official Review · Reviewer_9XQP · 2024-06-09
**Unit-Scaled Maximal Update Parametrization induces transferable hyperparameters in models while maintaining unit variance between passes, allowing for improved low-precision performance.**

**Confidence:** 3

**Summary:**

This paper introduces *Unit-Scaled Maximal Update Parametrization* (u-μP), which builds upon Maximal Update Parametrization (Yang & Hu, 2021) by combining it with the philosophy of unit variance between weights, activations, and gradients proposed in Unit Scaling (Blake et al., 2023). This allows for the hyperparameter transfer properties of μP to carry over to models that utilize lower precisions, such as FP8 casts.

To achieve this, the authors provide two main contributions over μP: modifying its original scaling scheme to follow unit scaling and simplifying the set of hyperparameters. Additionally, the authors provide per-operation scaling rules for Llama-specific architectural models to support their experimental results.

**Strengths:**

- The paper is well-written, with minimal typos and a clear motivation.
- The paper demonstrates strong empirical results in HP-transfer with FP8 models (See Figs 1., 4.)
- Notable reductions in loss are also demonstrated when compared to its predecessor μP.
- Figures are well-designed and easy to read.
- The introduction and background sections are concise yet informative, and give the reader a solid grasp of the resultant topics.
- The interpretability strengths of the method highlighted at the end of section 4 was interesting and informative of its strengths beyond experimental results.

**Weaknesses:**

Several things stand out,
- It is unclear sometimes whether or not a result is derived from an low or full-precision setting (eg. Fig 5), making statements like "Our u-μP scheme is more principled than that used for μP,..." harder to discern.
- Low-precision experiments are only carried out in FP8 settings. Perhaps it would be advantageous to see if the unit-variance properties proposed also result in improvements in 4bit or even 3bit models.
- To maintain unit variance in dot-product attention, the authors scale the pre-softmax product by $\alpha_{attn}$. However, the original scaled dot-product attention (in theory) already accounts for unit variance by multiplying the dot-product by $\sqrt{d}^{-1}$. It would be informative to specify why μP's choice of scaling by $d^{-1}$ rather than $\sqrt{d}^{-1}$ was kept over returning to the traditional formula.

**Limitations:**

- Past the scaling schematics outlined in Table 1, implementations of u-μP are model specific and therefore rely on unique scales for every operation in a particular architecture. This makes the primary contribution of u-μP more of a general practice rather than a specific algorithm, and thus results on additional models across different architectures could be beneficial.

**Suggestions:**

Overall, this was a well-written and interesting paper, and I'd like to thank the authors for this. Following my praises and critiques, some possible improvements would be:
- Experiments in lower-precisions (eg. 4bit or 3bit) models.
- Experiments across varying model architectures.
- There is a minor typo on line 313 (*it's* instead of *its*). It will be good to sweep the paper once and ensure simple things like these do not appear elsewhere.

---

### Official Review · Reviewer_TpYS · 2024-06-11
**Simpler, More Efficient Hyperparameter Sweeping**

**Confidence:** 4

**Summary:**

The paper presents a model parameterisation scheme called "u-µP", which combines the principles of Maximal Update Parameterisation (µP), a theoretical basis for optimal model parameterisation, with Unit Scaling, a weight scaling method original suited to low precision quantisation. Though appearing somewhat orthogonal in their use cases, the authors propose that these can be combined to create an effective model parameterisation, performing better than the original µP out of the box while retaining a key property of being able to transfer between models of different widths. This is done by combining the assumptions of both methods, to obtain a particular instance of µP parameterisation.

**Strengths:**

* Solid and well explained theoretical basis, building on recent research in model parameterisation and theories of model training.
* Evidently useful for anyone combining low precision (FP8) with hyperparameter sweeping, both popular and useful techniques.
* Interesting observations when comparing sweeping results between µP and u-µP, and good discussion of these differences.
* Experiments seem to suggest the scheme performs better with independent search, which is promising as this reduces expense.
* Appreciated level of detail regarding the implementations of scaling with different model layers.

**Weaknesses:**

* Somewhat limited evidence of u-µP's improvements over µP when it comes to full precision training. It is expected that the inclusion of Unit Scaling would improve performance in an FP8 context as shown; but I would be interested in more data to discern if there are further benefits.
* The authors chose to perform their experiments on the Llama architecture, which is a good choice, though it would be good to see how it applies on other models and datasets.
* It was observed that u-µP performs well in independent sweeping compared to µP, but there seems to be a lack of explanation or discussion on why this might be the case.
* Experiments while promising could be more thorough and compelling, in order to make clear the benefits of choosing u-µP for parameterisation.

---

### Official Review · Reviewer_cJ4s · 2024-06-12

**Confidence:** 4

**Summary:**

The paper presents a principled method to apply µP in conjunction with unit scaling (called u-µP). The paper showcases the benefits of merging the two independent strategies under a unified framework, enabling hyper-param transfer in low precision training settings. It further shows how to move from an abc-parameterization to absolute scales (which is needed for unit scaling), but can be done via the abc framework that µP enables. The paper finally identifies how moving to unit scaling can change some fundamentals, especially moving to more recent architectures such as Llama, and identify necessary fixes for those, specifically in the residual stream and the normalization blocks.

The authors provide detailed proofs for most of their proposed scaling methods. Based on the coherency of the presented proofs and results, I recommend an accept for the paper.

**Strengths:**

1. The paper identifies how to combine unit scaling and µP in a unified framework, enabling the properties of µP transfer for low-precision (FP8) training.
2. The paper presents principled proofs for most of the changes recommended to enable the combination of the ideas and delineates the base rules for the transfer clearly for practitioners in Table 1.
3. Through rigourous experiments, the authors show empirical validation for most of the proofs.
4. The authors combine the depth + width scaling for µP in this paper, which enables depth transfer even with unit scaling enabled (as shown in Figure 5). However, if we are to refer to the original depth scaling paper from Yang et al [1], it seems that ideally, transformers should not exhibit good depth transfer due to block_size >= 2 property of the blocks. How are the authors basing their assumptions on scaling with sqrt(base_depth)/depth for their transfer properties?
5. The paper gets rid of the dependency on base shapes - which often play a critical role in ensuring good transfer, but can also hinder transfer if the appropriate base widths are not considered.

[1] Tensor Programs VI: Feature Learning in Infinite-Depth Neural Networks (https://arxiv.org/abs/2310.02244)

**Weaknesses:**

1. While the authors recognize this, some of the math for the alpha_attn / alpha_silu is handwavy and is difficult to understand how the particular constants were arrived at.
2. The authors identify that the output layers in both the attn / ffn sub-blocks see large growth in weight / gradient magnitudes and propose moving to E5M2 to handle those layers. Wondering if following a principled way similar to Micikevicius et al. for E4M3 in the forward and E5M2 in the backward for all layers would be better?
3. While the intention for enabling simpler hyper-params + low-precision is understandable, there are too many things to account here for good transfer, making me wonder if there are reliable speedups to be gained from an implementation of the proposed method during implementation in actual runtime (not simulated like how the authors have currently done). Note that I'm not expecting the authors provide any concrete numbers for this - but even high-level projections will be useful to understand.

**Suggestions:**

A naming suggestion for the method: Since this is strictly not a parametrization anymore, it seems like calling it u-µP, which captures the general intent, is also slightly misleading. One recommendation is to potentially name the method as µnit Scaling or µS (muScaling) for short.

---

### Meta-Review · Area_Chair_3iEh · 2024-06-17

**Recommendation:** Accept (Poster)
**Confidence:** 4

**Metareview:**

This work proposes a new model parametrization scheme that combines Maximal Update Parametrization and Unit Scaling, combining the advantages of hyperparameter transfer and stable training in FP8 precision. Reviewers have appreciated the theoretical analysis of the method, rigorous experiments conducted to validate the approach, and the clarity of presentation. Most of the concerns expressed in reviews were relatively minor and refer mostly to additional analysis or clarifications of specific design choices, which can be done for the version of the work submitted to an archival venue. Thus, I recommend accepting this paper to the WANT workshop.

---

### Decision · Program_Chairs · 2024-06-18

**Decision:**

Accept (Poster)

**Comment:**

We thank the authors for their time and contribution to WANT and we are pleased to share that after the reviewing process the paper has been accepted. Congratulations! We encourage the authors to consider reviewers' feedback for the improvement of the camera-ready version. We hope to see you in person at the workshop and brainstorm on efficient training research together!